# High-probability bounds for Non-Convex Stochastic Optimization with Heavy Tails

**Ashok Cutkosky**
Boston University
ashok@cutkosky.com

**Harsh Mehta**
Google Research
harshm@google.com

## Abstract

We consider non-convex stochastic optimization using first-order algorithms for which the gradient estimates may have heavy tails. We show that a combination of gradient clipping, momentum, and normalized gradient descent yields convergence to critical points in high-probability with best-known rates for smooth losses when the gradients only have bounded $\mathfrak{p}$th moments for some $\mathfrak{p} \in (1, 2]$. We then consider the case of second-order smooth losses, which to our knowledge have not been studied in this setting, and again obtain high-probability bounds for any $\mathfrak{p}$. Moreover, our results hold for arbitrary smooth norms, in contrast to the typical SGD analysis which requires a Hilbert space norm. Further, we show that after a suitable "burn-in" period, the objective value will *monotonically decrease* whenever the current iterate is not a critical point, which provides intuition behind the popular practice of learning rate "warm-up" and also yields a last-iterate guarantee.

## 1 SGD and Heavy Tails

The standard algorithm for training large machine learning models today is stochastic gradient descent (SGD) and its myriad variants [13, 11, 19, 39, 28]. SGD enjoys many properties that help explain its empirical success: it can be implemented very easily, can efficiently process training data in a streaming manner, and even achieves the optimal convergence rate for finding critical points of smooth non-convex objectives [2]. In particular, SGD can be understood as optimizing the following stochastic optimization problem:

$$\min_{\vec{w}} F(\vec{w}) = \mathbb{E}_z[f(\vec{w}, z)]$$

where $f(\vec{w}, z)$ is some loss function taking model parameters $\vec{w}$ and a random example $z$, where $z$ has some distribution $P_z$. Then SGD consists of the update:

$$\vec{w}_{t+1} = \vec{w}_t - \eta \nabla f(\vec{w}_t, z_t)$$

where $z_1, \ldots, z_T$ are i.i.d. random variables distributed according to $P_z$, and $\eta$ is some real number called the learning rate. Assuming that $F$ is $L$-smooth and that $\nabla f(\vec{w}, z)$ satisfies $\mathbb{E}[\|\nabla f(\vec{w}, z)\|^2] \leq G^2$ for some $G$ for all $\vec{w}$, SGD satisfies [13]:

$$\frac{1}{T} \mathbb{E}\left[\sum_{t=1}^{T} \|\nabla F(\vec{w}_t)\|^2\right] \leq O\left(\frac{L\Delta}{T} + \frac{G^2}{\sqrt{T}}\right)$$

where $\Delta = F(\vec{w}_1) - \inf F(\vec{w})$. This implies that SGD takes $O(\epsilon^{-4})$ iterations to find an $\epsilon$-critical point, which is a point $\vec{w}$ such that $\|\nabla F(\vec{w})\| \leq \epsilon$ [13]. When the stochastic gradients are further assumed to have light sub-gaussian tails, [21] shows that SGD with momentum can achieve this same result in high probability as well.

However, recent work [30, 31, 42, 41, 43] suggests that the assumptions of bounded variance or light tails may be too optimistic in practical deep learning problems. For example, Zhang et al. [43] provides empirical evidence that large NLP models based on attention and transformers [35, 10] have *heavy-tailed* gradients. In particular, the variance is extremely large (perhaps even essentially infinite for practical purposes), but the $\mathfrak{p}th$ moment is bounded for some $\mathfrak{p} \in (1, 2]$. This setting is significantly more complicated, and SGD's convergence properties are difficult to characterize even when assuming slightly stronger tail bounds [29] or various versions of convexity [36].

A standard tool when working with heavy-tailed random variables is truncation, or *clipping*. Succinctly, if $X$ is a random variable such that $\mathbb{E}[\|X\|^2] = \infty$, $X$ itself may be poorly behaved due to the heavy tail, but a truncated value $X^{\text{clip}} = \frac{X}{|X|} \min(|X|, \tau)$ will be much more benign so long as $\tau$ is chosen in an appropriate manner [6, 23]. Fortuitously, gradient clipping has long been a standard technique to improve training of deep neural networks [26], and recent analysis [43, 14] suggests that clipping allows SGD to deal with heavy-tailed stochastic gradients. Specifically, [43] shows that SGD combined with clipping can find an $\epsilon$-critical point in expectation in $O\left(\epsilon^{-\frac{3\mathfrak{p}-2}{\mathfrak{p}-1}}\right)$ iterations.

We propose to improve upon this backdrop in three important ways. First, the prior analysis of [43] for non-convex convergence of SGD under heavy tails only provides an *in-expectation* analysis. In-expectation results are unsatisfying because the quantity measured for convergence (i.e. gradient norm or function value) is itself a random variable, and so might *also* have a heavy tail. In this case individual training runs could be quite poor. Since training modern networks is quite expensive [27, 5], it is important to be confident that each and every training run will produce a high-quality result. We provide a variant of the normalized SGD with momentum algorithm of [9] that incorporates gradient clipping and show that the method finds an $\epsilon$-critical point in $\tilde{O}\left(\epsilon^{-\frac{3\mathfrak{p}-2}{\mathfrak{p}-1}}\right)$ iterations with high probability.

Second, the vast majority of first-order optimization methods for stochastic non-convex optimization consider exclusively the standard $L_2$ or Hilbert-space norm. Such analyses can take advantage of the fact that $\mathbb{E}[\langle \nabla F(x), \vec{g} \rangle] = \|\nabla F(x)\|^2$ when $\vec{g}$ is an unbiased estimate of $\nabla F(x)$. In contrast, our algorithms find critical points when the gradient is measured with any smooth norm (i.e. in a Banach space). In this case, the gradient estimate $\vec{g}$ (which is a dual vector) must be converted into an appropriate primal vector. This operation is non-linear in general and so may add some bias. It turns out the key to operating with different norms is the popular practice of augmenting SGD iterates with momentum. When using momentum-based updates we do not rely on the same unbiasedness argument, and so the techniques can be extended to more general norms. This provides more concrete evidence of the theoretical advantages of momentum.

Third, recent theoretical advances in non-convex optimization have produced a number of new algorithms that avoid the lower-bounds of [2] by assuming extra structure on the objective $F$, such as *second-order smoothness*. In this case, [32, 1, 12, 9, 3] provide algorithms that achieve faster convergence rates. In particular, the algorithms of [12, 9] can find an $\epsilon$-critical point in $O(\epsilon^{3.5})$ iterations using a first-order stochastic oracle, which is the optimal rate [3]. To our knowledge, no analysis even in expectation exists for second-order smooth losses under heavy tails. We fill this gap, providing an algorithm that identifies an $\epsilon$-critical point in $\tilde{O}\left(\epsilon^{-\frac{5\mathfrak{p}-3}{2\mathfrak{p}-2}}\right)$ iterations with high probability. Further, in both smooth and second-order smooth cases we provide a convergence guarantee for the *last iterate* of the optimization algorithm: with high probability, the objective value for the final iterate will be smaller than the objective value at some approximate critical point.

Finally, we discuss an intriguing connection to the popular *warm-up* learning rate schedule [16, 15, 10], in which the learning rate starts small and increases over the first iterations of training. We show that for normalized SGD with momentum, after a brief "burn-in" period, the objective value decreases monotonically until an approximate critical point is reached. This provides an alternative motivation for warm-up: it would be reasonable to keep the learning rate small during the "burn-in" period. There is a long line of empirical research in understanding the need for warm-up. Warm-up has been shown to be connected to the use of layer-normalization, and that improved use of layer-norm can enable warm-up free training of transformer models [7, 37, 25]. Huang et al. [18] further build on these insights and proposes a weight initialization scheme which removes the need for both layer normalization and warm-up while training transformer models. Finally, Liu et al. [22] demonstrate that the variance of the learning rate in optimizers like Adam is abnormally high in the early stages of

the optimization process and further hypothesize that warm-up aids by reducing the variance in that stage. Our results provide a different view from an optimization perspective.

**Paper Organization:** in Section 1.1 we provide our concrete assumptions and setup. In Sections 2 and 3, we provide our algorithms and high-probability convergence bounds. In Section 4, we provide some empirical observations on the effect of the burn-in phase in our analysis.

## 1.1 Setup and assumptions

We assume that the objective $F(\vec{w}) = \mathbb{E}[f(\vec{w}, z)]$ is a Frechet-differentiable function mapping a Banach space $\mathcal{B}$ to the reals $\mathbb{R}$. Let $\|\cdot\|$ be the norm in $\mathcal{B}$ and $\|\cdot\|_\star$ be the dual norm in the dual space $\mathcal{B}^\star$. To preserve analogy and notation from the Hilbert space setting, we will write $\langle v, w \rangle$ to indicate the application of a dual vector $v \in \mathcal{B}^\star$ to a vector $w \in \mathcal{B}$. We assume that $\|\cdot\|_\star^p$ is Frechet differentiable and satisfies for some $p, C$ for all $x, y \in \mathcal{B}$:

$$\|x + y\|_\star^p \leq \|x\|_\star^p + \langle \nabla \|x\|_\star^p, y \rangle + C\|y\|_\star^p \tag{1}$$

Such a Banach space is called $(p, C)$-smooth. Note that if $\|\cdot\|$ is the standard $p$ norm in $\mathbb{R}^n$ for $p \in (1, 2]$, then $\|\cdot\|$ is the corresponding $q$ norm, and the dual space is $(2, \frac{1}{p-1})$-smooth. In the common case that $\mathcal{B}$ is a Hilbert space (e.g. $\mathbb{R}^n$ with the euclidean norm), then $\mathcal{B} = \mathcal{B}^\star$, $\|\cdot\| = \|\cdot\|_\star$ and $p = 2$ and $C = 1$. In this paper, we will exclusively consider the case $p = 2$ and $C \geq 1$. Finally, given any $v \in \mathcal{B}^\star$, we will write $d(v) \in \mathcal{B}$ to indicate a unit vector satisfying both $\|d(v)\| = 1$ and $\langle v, d(v) \rangle = \|v\|_\star$, which we assume exists for all $v$. For more background on Banach spaces, see Appendix E.

We assume $F$ is $L$-smooth, which means that $\|\nabla F(\vec{w}) - \nabla F(\vec{x})\|_\star \leq L\|\vec{w} - \vec{z}\|_\star$. In Section 3, we will also assume that $F$ is $\rho$-second-order smooth, which means that $\|(\nabla^2 F(\vec{w}) - \nabla^2 F(\vec{x}))v\|_\star \leq \rho\|\vec{w} - \vec{x}\|\|v\|$ for all $\vec{w}, \vec{x}, v \in \mathcal{B}$. We assume that $\nabla f(\vec{w}, z)$ satisfies both $\mathbb{E}[\|\nabla f(\vec{w}, z) - \nabla F(\vec{w})\|_\star^{\mathfrak{p}}] \leq G^{\mathfrak{p}}$ and $\mathbb{E}[\|\nabla f(\vec{w}, z)\|_\star^{\mathfrak{p}}] \leq G^{\mathfrak{p}}$ for some $G$ and some $\mathfrak{p} \in (1, 2]$. We will design algorithms such that after $T$ stochastic gradient evaluations, we can output a point $\vec{w}$ such that $\|\nabla F(\vec{w})\|_\star$ is small with high probability.

## 2 Normalized SGD with Momentum

In this section, we describe our algorithms for finding critical points with high probability in non-convex objectives. Our analysis makes use of gradient clipping, momentum, and normalized updates. Clipping is a standard technique for mitigating the effect of heavy tails, essentially by throwing out outlier data. Momentum is useful as it intuitively averages together many clipped gradients, resulting in a gradient estimate that concentrates about its mean with high probability. Normalizing the updates allows for significantly simplified analyses because the bias terms introduced by the moving average in momentum are now very precisely controlled by the learning rate, as originally analyzed for Hilbert spaces in [9]. In order to analyze this algorithm, we need the following critical Lemma, which characterizes the effect of taking a normalized SGD step (proved in appendix E).

**Lemma 1.** *Suppose $F : \mathcal{B} \to \mathbb{R}$ be a Frechet-differentiable $L$-Smooth function from a Banach space $\mathcal{B}$ to the reals. Let $w \in B$. Let $g^\star \in \mathcal{B}^\star$ and let $g \in \mathcal{B}$ be a unit-vector satisfying $\langle g^\star, g \rangle = \|g\|_\star$. Define $w' = w - \eta g$. Define $\epsilon = g^\star - \nabla F(w)$. Then:*

$$F(w') \leq F(w) - \eta\|\nabla F(w)\|_\star + 2\eta\|\epsilon\|_\star + \frac{L\eta^2}{2}$$

**Theorem 2.** *Suppose $\mathbb{E}_z[\|\nabla f(\vec{w}, z)\|_\star^{\mathfrak{p}}] \leq G^{\mathfrak{p}}$ for all $\vec{x}$ for some $G$. Suppose $F$ is $L$-smooth. Set $\beta = 1 - \alpha$, $\alpha = \frac{b}{T^{\frac{\mathfrak{p}}{3\mathfrak{p}-2}}}$ and $\eta = \frac{s}{T^{\frac{2\mathfrak{p}-1}{3\mathfrak{p}-2}}}$ for arbitrary constant $b$ and $s$ satisfying $\alpha \leq 1$ and set $\tau = G/\alpha^{1/\mathfrak{p}}$. Then with probability at least $1 - \delta$:*

$$\frac{1}{T}\sum_{t=1}^{T}\|\nabla F(\vec{w}_t)\|_\star \leq O\left(\frac{\log(T/\delta)}{T^{\frac{\mathfrak{p}-1}{3\mathfrak{p}-2}}}\right)$$

*where the big-Oh hides constant that depend on $L$, $G$ $b$, $s$, $C$ and $F(\vec{w}_1) - F(\vec{w}_{T+1})$, but not $T$ or $\delta$.*

---

**Algorithm 1** Normalized SGD with Clipping and Momentum

---

**Input:** Initial Point $\vec{w}_1$, learning rate $\eta$, momentum parameter $\beta$, clipping parameter $\tau$, time horizon $T$:
Set $\vec{m}_0 = 0$.
**for** $t = 1 \ldots T$ **do**
    Sample $z_t \sim P_z$.
    Set $\vec{g}_t^{\text{clip}} = \frac{\nabla f(\vec{w}_t, z_t)}{\|\nabla f(\vec{w}_t, z_t)\|_\star} \min(\tau, \|\nabla f(\vec{w}_1, z_t)\|_\star)$.
    Set $\vec{m}_t = \beta \vec{m}_{t-1} + (1 - \beta)\vec{g}_t^{\text{clip}}$.
    Set $\vec{w}_{t+1} = \vec{w}_t - \eta d(\vec{m}_t)$.
**end for**

---

This Theorem provides an analog of the standard in-expectation result for non-convex optimization, but now the result holds with high-probability. Note that this matches the in-expectation convergence rates from Zhang et al. [43] up to logarithmic factors. The next Theorem takes this theme one step further: it shows that after a brief "burn-in" period, the decrease in objective value will in fact be *monotonic* with high probability:

**Theorem 3.** *Under the assumptions of Theorem 2, define the constants:*

$$K = 10C \max(1, \log(3T/\delta)) + 4C^{1/2}\sqrt{\max(1, \log(3T/\delta))} + 1$$

$$Z = \frac{sL}{b} + GKb^{\frac{\mathfrak{p}-1}{\mathfrak{p}}}$$

*Then, with probability at least $1 - \delta$, for all $t$ satisfying:*

$$t \geq \mathcal{T} = \frac{T^{\frac{\mathfrak{p}}{3\mathfrak{p}-2}}}{b}\left(\frac{\mathfrak{p}-1}{3\mathfrak{p}-2}\log(T) + \log(G) - \log(Z)\right)$$

*we have*

$$\|\vec{m}_t - \nabla F(\vec{w}_t)\|_\star \leq \frac{2Z}{T^{\frac{\mathfrak{p}-1}{3\mathfrak{p}-2}}}$$

*Further, so long as*

$$\|\vec{m}_t\|_\star \geq 2\left(\frac{6Z}{T^{\frac{\mathfrak{p}-1}{3\mathfrak{p}-2}}} + \frac{Ls}{2T^{\frac{2\mathfrak{p}-1}{3\mathfrak{p}-2}}}\right) \tag{2}$$

*we have $F(\vec{w}_{t+1}) < F(\vec{w}_t) - \frac{\eta}{2}\|\vec{m}_t\|_\star$. Moreover, whenever (2) is not satisfied for $t \geq \mathcal{T}$, we must have*

$$\|\nabla F(\vec{w}_t)\|_\star \leq \frac{14Z}{T^{\frac{\mathfrak{p}-1}{3\mathfrak{p}-2}}} + \frac{Ls}{T^{\frac{2\mathfrak{p}-1}{3\mathfrak{p}-2}}}$$

*Finally, if $t$ is the iteration with smallest value of $\|\vec{m}_t\|_\star$ such that $t \geq \mathcal{T}$, we have:*

$$\|\nabla F(\vec{w}_t)\|_\star \leq O\left(\frac{\log(T/\delta)}{T^{\frac{\mathfrak{p}-1}{3\mathfrak{p}-2}}}\right)$$

Theorem 3 shows that after a small burn-in period $\mathcal{T} = o(T)$, $\vec{m}_t$ is very likely to be an extremely good estimate of the true gradient $\nabla F(\vec{x}_t)$. Moreover, we can empirically detect this event by monitoring the norm of the exponentially weighted moving average $\vec{m}$. This burn-in period is especially interesting given the success of learning rate schedules that incorporate a "warm-up" component, in which the learning rate actually increases during the first iterations. Our result seems to hint at a mechanism for this schedule: by keeping the learning rate small during the burn-in period, we can limit any increase in the objective value until we reach the stage at which it begins to decrease steadily.

Further, Theorem 3 provides a very concrete algorithm to find critical points: after the burn-in period of $\tilde{O}(T^{\frac{\mathfrak{p}}{3\mathfrak{p}-2}})$ iterates, we can simply return the point $\vec{x}_t$ with the smallest value of $\|\vec{m}_t\|$. Intuitively, this works because the objective value can only increase by a total of $O(T^{\frac{\mathfrak{p}}{3\mathfrak{p}-2}}\eta) = O(T^{\frac{1-\mathfrak{p}}{3\mathfrak{p}-2}}) = O(1)$ during the burn-in period so even without a learning rate "warm-up", we should not expect the burn-in period to have too significant an effect on the objective value. After the burn-in period, $\vec{m}_t$ is always very close to $\nabla F(\vec{w}_t)$, so we make steady progress until a critical point is encountered.

*Proof of Theorem 2.* Define $\hat{\epsilon}_t = \vec{m}_t - \nabla F(\vec{w}_t)$. and $\epsilon_t = \vec{g}_t^{\text{clip}} - \nabla F(\vec{w}_t)$. Also, define $S(a, b) = \nabla F(a) - \nabla F(b)$. By smoothness, we have $\|S(\vec{w}_t, \vec{w}_{t+1})\|_\star \leq L\|\vec{w}_t - \vec{w}_{t+1}\|_\star$

From the definition of $\vec{m}_{t+1}$, we have the following recursive formulation for any $t \geq 1$:

$$\vec{m}_{t+1} = (1 - \alpha)(\nabla F(\vec{w}_t) + \hat{\epsilon}_t) + \alpha \vec{g}_{t+1}^{\text{clip}}$$
$$= \nabla F(\vec{w}_{t+1}) + (1 - \alpha)(S(\vec{w}_t, \vec{w}_{t+1}) + \hat{\epsilon}_t) + \alpha \epsilon_{t+1}$$
$$\hat{\epsilon}_{t+1} = (1 - \alpha)S(\vec{w}_t, \vec{w}_{t+1}) + (1 - \alpha)\hat{\epsilon}_t + \alpha \epsilon_{t+1} \tag{3}$$

Further, by defining $\vec{m}_0 = 0$ and $\vec{w}_0 = \vec{w}_1$ so that $\hat{\epsilon}_0 = -\nabla F(\vec{w}_1)$, we have that (3) holds for $t = 0$ as well. Now, we unravel the recursion for all iterations:

$$\hat{\epsilon}_{t+1} = (1 - \alpha)^{t+1}\hat{\epsilon}_0 + \alpha \sum_{t'=0}^{t}(1 - \alpha)^{t'}\epsilon_{t+1-t'} + (1 - \alpha)\sum_{t'=0}^{t}(1 - \alpha)^{t'}S(\vec{w}_{t-t'}, \vec{w}_{t+1-t'})$$

Next, take the magnitude of both sides and use triangle inequality (using the fact that $\|S(\vec{w}_t, \vec{w}_{t+1})\|_\star \leq L\eta$):

$$\|\hat{\epsilon}_{t+1}\|_\star \leq (1 - \alpha)^{t+1}\|\hat{\epsilon}_0\|_\star + \alpha \left\|\sum_{t'=0}^{t}(1 - \alpha)^{t'}\epsilon_{t+1-t'}\right\|_\star + (1 - \alpha)\eta L \sum_{t'=0}^{t}(1 - \alpha)^{t'}$$

$$\leq (1 - \alpha)^{t+1}G + \frac{(1 - \alpha)\eta L}{\alpha} + \alpha \left\|\sum_{t'=0}^{t}(1 - \alpha)^{t'}\epsilon_{t+1-t'}\right\|_\star$$

where in the second line we have used $\vec{m}_0 = 0$ and $\|\nabla F(\vec{w}_0)\|_\star \leq G$, which follows since $\|\nabla F(\vec{w}_0)\|_\star \leq \mathbb{E}_z[\|\nabla f(\vec{w}_0, z)\|_\star^{\mathfrak{p}}]^{1/\mathfrak{p}} = G$.

Now, we invoke Lemma 9, a concentration inequality in Banach space (see appendix B). This implies that with probability at least $1 - \delta/T$:

$$\left\|\sum_{t'=0}^{t}(1 - \alpha)^{t'}\epsilon_{t+1-t'}\right\|_\star \leq 10C\tau \max(1, \log(3T/\delta)) + \sum_{t'=0}^{t}\frac{(1 - \alpha)^{t'}G^{\mathfrak{p}}}{\tau^{\mathfrak{p}-1}}$$
$$+ 4\left(C \sum_{t'=0}^{t}(1 - \alpha)^{2t'}G^{\mathfrak{p}}\tau^{2-\mathfrak{p}}\right)^{1/2}\sqrt{\max(1, \log(3T/\delta))}$$
$$\leq 10C\tau \max(1, \log(3T/\delta)) + \frac{G^{\mathfrak{p}}}{\alpha\tau^{\mathfrak{p}-1}} + \frac{4\left(CG^{\mathfrak{p}}\tau^{2-\mathfrak{p}}\right)^{1/2}\sqrt{\max(1, \log(3T/\delta))}}{\alpha^{1/2}}$$

Therefore, with probability at least $1 - \delta/T$:

$$\|\hat{\epsilon}_{t+1}\|_\star \leq (1 - \alpha)^{t+1}G + \frac{(1 - \alpha)\eta L}{\alpha} + 10C\alpha\tau \max(1, \log(3T/\delta))$$
$$+ 4\left(\alpha CG^{\mathfrak{p}}\tau^{2-\mathfrak{p}}\right)^{1/2}\sqrt{\max(1, \log(3T/\delta))} + \frac{G^{\mathfrak{p}}}{\tau^{\mathfrak{p}-1}}$$

Now, with $\tau = \frac{G}{\alpha^{1/\mathfrak{p}}}$, and $D_\delta = \max(1, \log(3T/\delta))$ this becomes:

$$\leq (1 - \alpha)^{t+1}G + \frac{(1 - \alpha)\eta L}{\alpha} + G\alpha^{1-1/\mathfrak{p}}\left[10CD_\delta + 4C^{1/\mathfrak{p}}\sqrt{D_\delta} + 1\right]$$
$$= (1 - \alpha)^{t+1}G + \frac{(1 - \alpha)\eta L}{\alpha} + G\alpha^{1-1/\mathfrak{p}}K \tag{4}$$

where we define the constant $K = 10CD_\delta + 4C^{1/2}\sqrt{D_\delta} + 1$.

Now, we have from Lemma 1:

$$\sum_{t=1}^{T}\|\nabla F(\vec{w}_t)\|_\star \leq \frac{F(\vec{w}_1) - F(\vec{w}_T)}{\eta} + \frac{LT\eta}{2} + 2\sum_{t=1}^{T}\|\hat{\epsilon}_t\|_\star$$

so with probability at least $1 - \delta$:

$$\leq \frac{F(\vec{w}_1) - F(\vec{w}_T)}{\eta} + \frac{LT\eta}{2} + 2\sum_{t=1}^{T}(1-\alpha)^t G + \frac{(1-\alpha)\eta L}{\alpha} + G\alpha^{1-1/\mathfrak{p}}K$$

$$\leq \frac{F(\vec{w}_1) - F(\vec{w}_T)}{\eta} + \frac{LT\eta}{2} + \frac{2G(1-\alpha)}{\alpha} + \frac{2(1-\alpha)\eta TL}{\alpha} + 2GT\alpha^{1-1/\mathfrak{p}}K$$

Now set $\alpha = \frac{b}{T^{\frac{\mathfrak{p}}{3\mathfrak{p}-2}}}$ and $\eta = \frac{s}{T^{\frac{2\mathfrak{p}-1}{3\mathfrak{p}-2}}}$. This yields:

$$\frac{1}{T}\sum_{t=1}^{T}\|\nabla F(\vec{w}_t)\| \leq O\left(T^{\frac{1-\mathfrak{p}}{3\mathfrak{p}-2}}\log(T/\delta)\right)$$

$\square$

The proof of Theorem 3 follows essentially the same idea as that of Theorem 2. However, this time we use the explicit bound on $\|\hat{\epsilon}_t\|_\star$ and observe that after the burn-in period, the contribution from $\|\epsilon_1\|_\star$ has decreased exponentially to be insignificant, and so $\vec{m}_t$ will be a very high-quality estimate of $\nabla F(\vec{x}_t)$ at every single iteration.

*Proof of Theorem 3.* Use the settings $\eta = \frac{s}{T^{\frac{2\mathfrak{p}-1}{3\mathfrak{p}-2}}}$ and $\alpha = \frac{b}{T^{\frac{\mathfrak{p}}{3\mathfrak{p}-2}}}$ and notice that from equation (4) that with probability $1 - \delta$, for all $t$:

$$\|\hat{\epsilon}_{t+1}\|_\star \leq (1-\alpha)^t G + \frac{sL}{bT^{\frac{\mathfrak{p}-1}{3\mathfrak{p}-2}}} + \frac{GKb^{\frac{\mathfrak{p}-1}{\mathfrak{p}}}}{T^{\frac{\mathfrak{p}-1}{3\mathfrak{p}-2}}} = (1-\alpha)^t G + \frac{Z}{T^{\frac{\mathfrak{p}-1}{3\mathfrak{p}-2}}}$$

Further, we have

$$\log(1-\alpha) \leq -\alpha$$
$$\log\left(G(1-\alpha)^t\right) \leq -t\alpha + \log(G)$$

Therefore, if $t \geq \mathcal{T}$, then we have with probability at least $1 - \delta$:

$$\|\hat{\epsilon}_{t+1}\|_\star \leq \frac{2Z}{T^{\frac{\mathfrak{p}-1}{3\mathfrak{p}-2}}}$$

This implies the bound on $\|\nabla F(\vec{w}_t)\|_\star$ when $\|\vec{m}_t\|_\star$ is smaller than the threshold described in the Theorem statement. Next, again from Lemma 1,

$$F(\vec{w}_{t+1}) \leq F(\vec{w}_t) - \eta\|\nabla F(\vec{w}_t)\|_\star + 2\eta\|\hat{\epsilon}_t\|_\star + \frac{L\eta^2}{2}$$

$$\leq F(\vec{w}_t) - \eta\|\vec{m}_t\|_\star + 3\eta\|\hat{\epsilon}_t\|_\star + \frac{L\eta^2}{2}$$

$$\leq F(\vec{w}_t) - \eta\|\vec{m}_t\|_\star + \eta\left(\frac{6Z}{T^{\frac{\mathfrak{p}-1}{3\mathfrak{p}-2}}} + \frac{Ls}{2T^{\frac{2\mathfrak{p}-1}{3\mathfrak{p}-2}}}\right)$$

from which we can conclude that if $\|\vec{m}_t\|_\star \geq 2\left(\frac{6Z}{T^{\frac{\mathfrak{p}-1}{3\mathfrak{p}-2}}} + \frac{Ls}{2T^{\frac{2\mathfrak{p}-1}{3\mathfrak{p}-2}}}\right)$, $F(\vec{w}_{t+1}) \leq F(\vec{w}_t) - \frac{\eta}{2}\|\vec{m}_t\|_\star$.

Now, for the final part of the Theorem, observe that since $F$ is $G$-Lipschitz, we must have that

$$F(\vec{w}_\mathcal{T}) - F(\vec{w}_1) \leq G\mathcal{T}\eta = O(\log(T))$$

so that $F(\vec{w}_\mathcal{T}) - F(\vec{w}_{T+1}) \leq O(F(\vec{w}_1) - F(\vec{w}_{T+1}) + \log(T))$. Now consider two cases, either

$$\min_{t \geq \mathcal{T}}\|\vec{m}_t\|_\star \leq 2\left(\frac{6Z}{T^{\frac{\mathfrak{p}-1}{3\mathfrak{p}-2}}} + \frac{Ls}{2T^{\frac{2\mathfrak{p}-1}{3\mathfrak{p}-2}}}\right)$$

or not. In the first case, by our bound on $\|\hat{\epsilon}_t\|_\star$, the desired bound on $\|\nabla F(\vec{w}_t)\|_\star$ follows. In the latter case, we have

$$F(\vec{w}_{T+1}) \leq F(\vec{w}_\mathcal{T}) - \frac{\eta}{2}\sum_{t=\mathcal{T}}^{T}\|\vec{m}_t\|_\star$$

So that

$$\min_{t \geq \mathcal{T}} \|\vec{m}_t\|_\star \leq O\left(\frac{F(\vec{w}_1) - F(\vec{w}_{T+1}) + \log(T)}{T\eta}\right) = \tilde{O}\left(\frac{1}{T^{\frac{2p-1}{3p-2}}}\right)$$

And so again the result follows from our bound on $\|\hat{\epsilon}_t\|_\star$. $\qquad\square$

In addition to providing a method for identifying critical points, Theorem 3 also provides an intuitively desirable guarantee about the *last-iterate* of the algorithm $\vec{w}_T$. Specifically, one can easily check that the following Corollary:

**Corollary 4.** *Under the notation and assumptions of Theorem 3, with probability at least $1 - \delta$, there exists some $\hat{w}$ such that both of the following inequalities hold:*

$$\|\nabla F(\hat{w})\|_\star \leq O\left(\frac{\log(T/\delta)}{T^{\frac{p-1}{3p-2}}}\right)$$
$$F(\vec{w}_T) \leq F(\hat{w})$$

*where $\vec{w}_T$ is the last iterate of Algorithm 1*

*Proof.* Set $\hat{t}$ to be the last index such that $\vec{w}_{\hat{t}}$ does not satisfy (2). Set $\hat{w} = \vec{w}_{\hat{t}}$. Then Theorem 3 implies the result. $\qquad\square$

In most applications we are not actually interested in finding critical points, but instead wish to actually minimize the objective $F$. This observation tells us that our objective value is at least never worse than it would have been if had indeed searched specifically for a critical point. Moreover, by providing a guarantee about $\vec{w}_T$, we have a closer match to practical use-cases: many of the theoretical analyses of non-convex stochastic gradient methods [13, 38, 20] only show that a *randomly selected iterate* has a small gradient norm. In contrast, Corollary 4 provides a non-asymptotic guarantee for the final iterate, which is the iterate most likely to be deployed after training in practice.

## 3 Second-Order Smooth Losses

In this section, we provide our extension to losses that are *second order smooth*. The overall method is very similar: we employ normalized SGD with momentum. However, in order to take advantage of the second-order smoothness, we will need to use a more advanced form of momentum. Specifically, we use implicit gradient transport [4], as implemented by [9] in their NIGT algorithm. This algorithm replaces the standard momentum update with an extrapolation procedure:

$$\vec{m}_t = \beta\vec{m}_{t-1} + (1 - \beta)\nabla f\left(\vec{w}_t + \frac{\beta(\vec{w}_t - \vec{w}_{t-1})}{1 - \beta}, z_t\right)$$

By evaluating the gradient at the "shifted" point $\vec{w}_t + \frac{\beta(\vec{w}_t - \vec{w}_{t-1})}{1-\beta}$ and performing a second-order Taylor expansion enabled by the second-order smoothness assumption, it is possible to show that $\vec{m}_t$ is a less-biased estimate of $\nabla F(\vec{w}_t)$ than it would be using the standard momentum update. We augment this procedure with gradient clipping in Algorithm 2 below, and provide its analysis in Theorems 5 and 6, which are directly analogous to Theorems 2 and 3.

**Theorem 5.** *Suppose $\mathbb{E}_z[\|\nabla f(\vec{w}, z)\|^p] \leq G^p$ for all $\vec{w}$ for some $G$. Suppose $F$ is L-smooth and $\rho$-second-order smooth. Set $\beta = 1 - \alpha$, $\alpha = \frac{b}{T^{\frac{2p}{5p-3}}}$ and $\eta = \frac{a}{T^{\frac{3p-1}{5p-3}}}$ for arbitrary constants $b$ and $c$ satisfying $\alpha \leq 1$, and set $\tau = \frac{G}{\alpha^{1/p}}$. Then with probability at least $1 - \delta$:*

$$\frac{1}{T}\sum_{t=1}^{T}\|\nabla F(\vec{w}_t)\|_\star \leq O\left(\frac{\log(T/\delta)}{T^{\frac{2p-2}{5p-3}}}\right)$$

*where the big-Oh hides constant that depend on $L$, $\rho$, $G$, $b$, $s$, $C$, and $F(\vec{w}_1) - F(\vec{w}_{T+1})$, but not $T$ or $\delta$.*

We also have a direct analog of Theorem 3

**Algorithm 2** NIGT with Clipping

---

**Input:** Initial Point $\vec{w}_1$, learning rate $\eta$, momentum parameter $\beta$, clipping parameter $\tau$, time horizon $T$:
Set $\vec{m}_0 = 0$.
**for** $t = 1 \ldots T$ **do**
    Sample $z_t \sim P_z$.
    $\vec{x}_t = \vec{w}_t + \frac{\beta(\vec{w}_t - \vec{w}_{t-1})}{1-\beta}$
    Set $\vec{g}_t^{\text{clip}} = \frac{\nabla f(\vec{x}_t, z_t)}{\|\nabla f(\vec{x}_t, z_t)\|_\star} \min(\tau, \|\nabla f(\vec{x}_1, z_t)\|_\star)$.
    Set $\vec{m}_t = \beta \vec{m}_{t-1} + (1-\beta)\vec{g}_t^{\text{clip}}$.
    Set $\vec{w}_{t+1} = \vec{w}_t - \eta d(\vec{m}_t)$.
**end for**

---

**Theorem 6.** *Under the assumptions of Theorem 5, define the constants:*

$$K = 10 \max(1, \log(3T/\delta)) + 4\sqrt{C \max(1, \log(3T/\delta))} + 1$$

$$Z = \frac{\rho s^2}{b^2} + GKb^{\frac{\mathfrak{p}-1}{\mathfrak{p}}}$$

*Then, with probability at least $1 - \delta$, if*

$$t \geq \mathcal{T} = \frac{T^{\frac{2\mathfrak{p}}{5\mathfrak{p}-3}}}{b}\left(\frac{2\mathfrak{p}-2}{5\mathfrak{p}-3}\log(T) + \log(G) - \log(Z)\right)$$

*Then we have*

$$\|\vec{m}_t - \nabla F(\vec{x}_t)\| \leq \frac{2Z}{T^{\frac{2\mathfrak{p}-2}{5\mathfrak{p}-3}}}$$

*And so long as*

$$\|\vec{m}_t\|_\star \geq 2\left(\frac{6Z}{T^{\frac{2\mathfrak{p}-2}{5\mathfrak{p}-3}}} + \frac{Ls}{2T^{\frac{3\mathfrak{p}-1}{5\mathfrak{p}-3}}}\right) \tag{5}$$

*we have $F(\vec{w}_{t+1}) < F(\vec{w}_t) - \frac{\eta}{2}\|\vec{m}_t\|$. Moreover, if (5) is ever not satisfied, we must have*

$$\|\nabla F(\vec{w}_t)\|_\star \leq \frac{14Z}{T^{\frac{2\mathfrak{p}-2}{5\mathfrak{p}-3}}} + \frac{Ls}{T^{\frac{3\mathfrak{p}-1}{5\mathfrak{p}-3}}}$$

*Finally, if $t$ is the iteration with smallest value of $\|\vec{m}_t\|$ such that $t \geq \mathcal{T}$, we have with probability at least $1 - \delta$:*

$$\|\nabla F(\vec{w}_t)\|_\star \leq O\left(\frac{\log(T/\delta)}{T^{\frac{2\mathfrak{p}-2}{5\mathfrak{p}-3}}}\right)$$

Finally, we note that we have a statement about the last iterate of Algorithm 2 that is directly analogous to Corollary 4:

**Corollary 7.** *Under the notation and assumptions of Theorem 6, with probability at least $1 - \delta$, there exists some $\hat{w}$ such that both of the following inequalities hold:*

$$\|\nabla F(\hat{w})\|_\star \leq O\left(\frac{\log(T/\delta)}{T^{\frac{2\mathfrak{p}-2}{5\mathfrak{p}-2}}}\right)$$

$$F(\vec{w}_T) \leq F(\hat{w})$$

*where $\vec{w}_T$ is the last iterate of Algorithm 2*

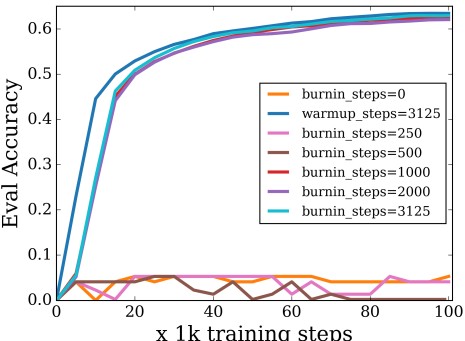 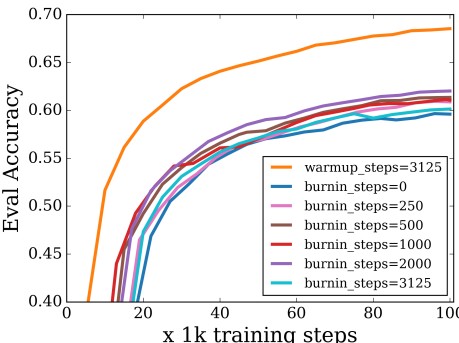

Figure 1: Comparison between warm-up and burn-in strategies on Masked language modeling BERT-Large pre-training task. Plot (a) shows the eval accuracy when trained with Algorithm 1 and plot (b) does the same when trained with LAMB.

## 4 Experiments

Our theoretical analysis suggests that there is a short "burn-in" phase in our algorithms during which the momentum estimate $\vec{m}_t$ may be far from the true gradient $\nabla F(\vec{w}_t)$, and after this period, $\vec{m}_t$ will always be close to $\nabla F(\vec{w}_t)$. As a result, during the burn-in phase it is possible for the optimizer to make slow or even negative progress, while afterwards we should expect more steady improvement. This suggests that one might set the learning rate $\eta = 0$ throughout the burn-in phase and only begin taking steps after $\vec{m}_t$ has accumulated enough gradients to be come a good estimate. This procedure is reminiscent of the common practice of learning rate "warm-up", in which the learning rate starts at 0 and is linearly increased to some desired value during the initial phase of training, and then later decreased. Thus, we sought to answer how much of the success of the warm-up schedule can be attributed to burn-in.

To answer this question, we conducted an experimental study on the BERT pretraining task. We chose to experiment on BERT since it has been shown to have heavy tails empirically [43] and the common practice is already to clip the gradients. Figure 3 plots the masked language model accuracy of BERT-Large model when trained with Algorithm 1. We also include experimental results using the LAMB optimizer instead of Algorithm 1. LAMB optimizer has been shown to obtain state of the art performance on BERT task [40]. Moreover, both our analysis and the LAMB optimizer relies on normalized updates. Note that we also include experimental results using the smaller BERT-Base model in the appendix. All our experiments were conducted using the Tensorflow framework on TPUv3 architecture.

When using Algorithm 1, we employ base learning rate $\eta_0$ of 0.3 and batch size of 512. We came up with this learning rate by performing grid search $\eta_0 \in [1.0, 0.5, 0.1, 0.05, 0.01, 0.005, 0.001, 0.0005, 0.0001]$ and choosing the one which attains the best eval accuracy. To obtain the optimal base learning rate of 0.3, we again ran a grid search $\eta_0 \in [0.1, 0.2, 0.3, 0.4, 0.5]$. Our warmup baseline employs the standard practice of linear warm-up for 3125 steps and polynomical decay of $\eta_t = \eta_0 * (1 - \frac{t}{T})$ for the rest of the steps. For a fair comparison with the baseline, we start the polynomial decay after 3125 steps regardless of how many steps were used for burn-in. We set the learning rate to the constant base learning rate value of 0.3 for BERT-Large for steps in between burn-in and polynomial decay. For the models trained using burn-in, instead of linear warm-up, we set the base learning rate $\eta_0$ to zero for the amount of steps we run the burn-in for. As shown in Fig 1, we perform a grid search across $[0, 250, 500, 1000, 2000, 3125]$ steps for the initial burn-in period and compare their results. Notice that, without warm-up or burn-in (i.e. 0 burn-in steps), we see very poor performance. Further, this poor performance persists for smaller values of burn-in, but with large enough burn-in values we suddenly see very good results. Once this critical level of burn-in is achieved, we continue to see small gains as the amount of burn-in is increased. This suggests that without burn-in or warm-up, the first iterations are taking extremely poor steps in the non-convex landscape that may destroy some favorable properties of the initialization point. With enough burn-in, each step is well-aligned with the true gradient and so we may avoid this bad behavior. Further, we see that employing burn-in recovers almost all of the advantage gained

from employing warm-up (final accuracy of 62.95% with burn-in vs. 63.45% with warm-up). These results lend empirical support to our theoretical intuition that the initial steps of the optimization may be unhelpful, or even hurtful.

For the LAMB optimizer, we reproduce all the hyperparameters from You et al. [40]. Most importantly, we train all our models using the base learning rate $\eta_0$ of 6.25e-4 and batch size of 512. As shown in Fig 1, again the warmup baseline employs the standard practice of linear warm-up for 3125 steps [40] and polynomical decay of $\eta_t = \eta_0 * (1 - \frac{t}{T})$ for the rest of the steps. Again, for a fair comparison with the baseline, we start the polynomial decay after 3125 steps regardless of how many steps were used for burn-in. We set the learning rate to the constant base learning rate value of 6.25e-4 for steps in between burn-in and polynomial decay. Note that when using the LAMB optimizer, warm-up baseline does considerably better than models where burn-in was employed. However, notice that the gap between employing warm-up and using *neither* warm-up nor burn-in is considerably smaller than with Algorithm 1, for which not using either technique results in extremely poor accuracy. This suggests that the additional heuristics employed by the LAMB optimizer over the simple normalized SGD approach of Algorithm 1 are providing some non-trivial robustness that remains to be understood. Further, there is still a noticeable gap between the linear learning-rate increase in warm-up and the coarser burn-in policy. It remains to be seen exactly what mechanism this smoother policy employs to aid optimization.

## 5 Conclusion

We have provided algorithms based on normalized SGD with momentum and gradient clipping that provide high probability-rates for finding critical points of non-convex objectives with heavy-tailed gradients. We also show that as a by-product of our high-probability bounds, the algorithms provide a pleasing result for the *last iterate* of the optimization procedure. Specifically, the last iterate has function value smaller than the function value of a critical point.

Finally, we identify a critical "burn-in" phase early in optimization in which the optimizer may be making poor decisions. Empirically, we show that keeping the learning rate zero for the first iterations indeed does help training compared to simply running the optimizer with no special treatment during this phase. It is an interesting question to precisely identify what other advantages are provided by the increasing schedule used by warm-up during this phase.

**Acknowledgments:** The authors thank the anonymous reviewers for many helpful suggestions. AC was a visiting researcher at Google when this work was completed.

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
