# A  Proofs for Section 3

In this section we provide the missing proofs of Theorems 5 and 6.

**Theorem 5.** *Suppose* $\mathbb{E}_z[\|\nabla f(\vec{w}, z)\|^{\mathfrak{p}}] \leq G^{\mathfrak{p}}$ *for all* $\vec{w}$ *for some* $G$. *Suppose* $F$ *is L-smooth and* $\rho$-*second-order smooth. Set* $\beta = 1 - \alpha$, $\alpha = \frac{b}{T^{\frac{2\mathfrak{p}}{5\mathfrak{p}-3}}}$ *and* $\eta = \frac{a}{T^{\frac{3\mathfrak{p}-1}{5\mathfrak{p}-3}}}$ *for arbitrary constants* $b$ *and* $c$ *satisfying* $\alpha \leq 1$, *and set* $\tau = \frac{G}{\alpha^{1/\mathfrak{p}}}$. *Then with probability at least* $1 - \delta$:

$$\frac{1}{T} \sum_{t=1}^{T} \|\nabla F(\vec{w}_t)\|_{\star} \leq O\left(\frac{\log(T/\delta)}{T^{\frac{2\mathfrak{p}-2}{5\mathfrak{p}-3}}}\right)$$

*where the big-Oh hides constant that depend on* $L$, $\rho$, $G$, $b$, $s$, $C$, *and* $F(\vec{w}_1) - F(\vec{w}_{T+1})$, *but not* $T$ *or* $\delta$.

*Proof.* The start of our proof is very similar to that of Theorem 3 in [9]. Similar to the proof of Theorem 2, define $\hat{\epsilon}_t = \vec{m}_t - \nabla F(\vec{w}_t)$ and $\epsilon_t = \vec{g}_t^{\text{clip}} - \nabla F(\vec{x}_t)$. Also (and as a departure from Theorem 2), define $S(a, b) = \nabla F(a) - \nabla F(b) - \nabla^2 F(b)(a - b)$. Then we have a recursion for all $t \geq 1$:

$$\vec{m}_{t+1} = (1 - \alpha)(\nabla F(\vec{w}_t) + \hat{\epsilon}_t) + \alpha \vec{g}_{t+1}^{\text{clip}}$$
$$= (1 - \alpha)(\nabla F(\vec{w}_{t+1}) + \nabla^2 F(\vec{w}_{t+1})(\vec{w}_t - \vec{w}_{t+1})) + (1 - \alpha)(S(\vec{w}_t, \vec{w}_{t+1}) + \hat{\epsilon}_t)$$
$$+ \alpha(\nabla F(\vec{w}_{t+1}) + \nabla^2 F(\vec{w}_{t+1})(\vec{x}_{t+1} - \vec{w}_{t+1})) + \alpha(S(\vec{x}_{t+1}, \vec{w}_{t+1}) + \epsilon_{t+1})$$

Now, observe that by definition of $\vec{x}_{t+1}$, the hessian-dependent terms cancel:

$$= \nabla F(\vec{w}_{t+1}) + (1 - \alpha)(S(\vec{w}_t, \vec{w}_{t+1}) + \hat{\epsilon}_t) + \alpha(S(\vec{x}_{t+1}, \vec{w}_{t+1}) + \epsilon_{t+1})$$
$$\hat{\epsilon}_{t+1} = (1 - \alpha)(S(\vec{w}_t, \vec{w}_{t+1}) + \hat{\epsilon}_t) + \alpha(S(\vec{x}_{t+1}, \vec{w}_{t+1}) + \epsilon_{t+1})$$

Further, as in the proof of Theorem 2, we set $\vec{w}_0 = \vec{w}_1$ and notice $\hat{\epsilon}_0 = -\nabla F(\vec{w}_1)$ so that the above recursion holds for $t = 0$ as well. Then we again unwind the recursion:

$$\hat{\epsilon}_{t+1} = (1 - \alpha)^{t+1}\hat{\epsilon}_0 + \alpha \sum_{t'=0}^{t}(1 - \alpha)^{t'}\epsilon_{t+1-t'} + (1 - \alpha)\sum_{t'=0}^{t}(1 - \alpha)^{t'} S(\vec{w}_{t-t'}, \vec{w}_{t-t'+1})$$

$$+ \alpha \sum_{t'=0}^{t}(1 - \alpha)^{t'} S(\vec{x}_{t-t'+1}, \vec{w}_{t-t'+1})$$

$$\|\hat{\epsilon}_{t+1}\|_{\star} \leq (1 - \alpha)^{t+1}\|\hat{\epsilon}_0\|_{\star} + \alpha \left\|\sum_{t'=0}^{t}(1 - \alpha)^{t'}\epsilon_{t+1-t'}\right\|_{\star} + (1 - \alpha)\left\|\sum_{t'=0}^{t}(1 - \alpha)^{t'} S(\vec{w}_{t-t'}, \vec{w}_{t-t'+1})\right\|_{\star}$$

$$+ \alpha \left\|\sum_{t'=0}^{t}(1 - \alpha)^{t'} S(\vec{x}_{t-t'+1}, \vec{w}_{t-t'+1})\right\|_{\star}$$

Now, by second-order smoothness, we have $\|S(a, b)\|_{\star} \leq \frac{\rho}{2}\|a - b\|^2$ (see Proposition 17) and further $\frac{(1-\alpha)^2}{\alpha} + 1 - \alpha \leq 2\frac{1-\alpha}{\alpha}$, so that combining the last two sums yields:

$$\leq (1 - \alpha)^{t+1}\|\hat{\epsilon}_0\|_{\star} + \alpha \left\|\sum_{t'=0}^{t}(1 - \alpha)^{t'}\epsilon_{t+1-t'}\right\|_{\star} + \frac{(1 - \alpha)\rho\eta^2}{\alpha}\sum_{t'=0}^{t}(1 - \alpha)^{t'}$$

$$\leq (1 - \alpha)^{t+1}\|\hat{\epsilon}_0\|_{\star} + \alpha \left\|\sum_{t'=0}^{t}(1 - \alpha)^{t'}\epsilon_{t+1-t'}\right\|_{\star} + \frac{(1 - \alpha)\rho\eta^2}{\alpha^2}$$

Now, by exactly the same argument as in Theorem 2, defining $D_\delta = \max(1, \log(1/\delta))$ we have with probability at least $1 - \delta/T$:

$$\|\hat{\epsilon}_{t+1}\|_{\star} \leq (1 - \alpha)^{t+1}G + \frac{(1 - \alpha)\rho\eta^2}{\alpha^2} + 10C\alpha\tau D_\delta + 4\sqrt{\alpha CG^{\mathfrak{p}}\tau^{2-\mathfrak{p}}D_\delta} + \frac{G^{\mathfrak{p}}}{\tau^{\mathfrak{p}-1}}$$

Then, with $\tau = \frac{G}{\alpha^{1/\mathfrak{p}}}$, this becomes:

$$\leq (1-\alpha)^{t+1} G + \frac{(1-\alpha)\rho\eta^2}{\alpha^2} + G\alpha^{1-1/\mathfrak{p}} \left[ 10CD_\delta + 4\sqrt{CD_\delta} + 1 \right]$$

$$= (1-\alpha)^{t+1} G + \frac{(1-\alpha)\rho\eta^2}{\alpha^2} + G\alpha^{1-1/\mathfrak{p}} K \tag{6}$$

where we again define the constant $K = 10D_\delta + 4\sqrt{CD_\delta} + 1$.

Now, we have from Lemma 1:

$$\sum_{t=1}^{T} \|\nabla F(\vec{w}_t)\|_\star \leq \frac{F(\vec{w}_1) - F(\vec{w}_T)}{\eta} + \frac{LT\eta}{2} + 2\sum_{t=1}^{T} \|\hat{\epsilon}_t\|_\star$$

so with probability at least $1 - \delta$:

$$\leq \frac{F(\vec{w}_1) - F(\vec{w}_T)}{\eta} + \frac{LT\eta}{2} + 2\sum_{t=1}^{T} \left[ (1-\alpha)^{t+1} G + \frac{(1-\alpha)\rho\eta^2}{\alpha^2} + G\alpha^{1-1/\mathfrak{p}} K \right]$$

$$\leq \frac{F(\vec{w}_1) - F(\vec{w}_T)}{\eta} + \frac{LT\eta}{2} + \frac{2G(1-\alpha)}{\alpha} + \frac{2(1-\alpha)T\rho\eta^2}{\alpha^2} + 2GT\alpha^{1-1/\mathfrak{p}} K$$

Now set $\alpha = \frac{b}{T^{\frac{2\mathfrak{p}}{5\mathfrak{p}-3}}}$ and $\eta = \frac{s}{T^{\frac{3\mathfrak{p}-1}{5\mathfrak{p}-3}}}$. This yields with probability at least $1 - \delta$:

$$\frac{1}{T} \sum_{t=1}^{T} \|\nabla F(\vec{w}_t)\|_\star \leq O\left( T^{\frac{2-2\mathfrak{p}}{5\mathfrak{p}-3}} \log(T/\delta) \right)$$

$\square$

**Theorem 6.** *Under the assumptions of Theorem 5, define the constants:*

$$K = 10\max(1, \log(3T/\delta)) + 4\sqrt{C\max(1, \log(3T/\delta))} + 1$$

$$Z = \frac{\rho s^2}{b^2} + GKb^{\frac{\mathfrak{p}-1}{\mathfrak{p}}}$$

*Then, with probability at least $1 - \delta$, if*

$$t \geq \mathcal{T} = \frac{T^{\frac{2\mathfrak{p}}{5\mathfrak{p}-3}}}{b} \left( \frac{2\mathfrak{p}-2}{5\mathfrak{p}-3} \log(T) + \log(G) - \log(Z) \right)$$

*Then we have*

$$\|\vec{m}_t - \nabla F(\vec{x}_t)\| \leq \frac{2Z}{T^{\frac{2\mathfrak{p}-2}{5\mathfrak{p}-3}}}$$

*And so long as*

$$\|\vec{m}_t\|_\star \geq 2\left( \frac{6Z}{T^{\frac{2\mathfrak{p}-2}{5\mathfrak{p}-3}}} + \frac{Ls}{2T^{\frac{3\mathfrak{p}-1}{5\mathfrak{p}-3}}} \right) \tag{5}$$

*we have $F(\vec{w}_{t+1}) < F(\vec{w}_t) - \frac{\eta}{2}\|\vec{m}_t\|$. Moreover, if (5) is ever not satisfied, we must have*

$$\|\nabla F(\vec{w}_t)\|_\star \leq \frac{14Z}{T^{\frac{2\mathfrak{p}-2}{5\mathfrak{p}-3}}} + \frac{Ls}{T^{\frac{3\mathfrak{p}-1}{5\mathfrak{p}-3}}}$$

*Finally, if $t$ is the iteration with smallest value of $\|\vec{m}_t\|$ such that $t \geq \mathcal{T}$, we have with probability at least $1 - \delta$:*

$$\|\nabla F(\vec{w}_t)\|_\star \leq O\left( \frac{\log(T/\delta)}{T^{\frac{2\mathfrak{p}-2}{5\mathfrak{p}-3}}} \right)$$

*Proof.* Use the settings $\eta = \frac{s}{T^{\frac{3\mathfrak{p}-1}{5\mathfrak{p}-3}}}$ and $\alpha = \frac{b}{T^{\frac{2\mathfrak{p}}{5\mathfrak{p}-3}}}$ and notice that from equation (6) that with probability $1-\delta$, for all $t$:

$$\|\hat{\epsilon}_{t+1}\|_\star \leq (1-\alpha)^t G + \frac{(1-\alpha)\rho\eta^2}{\alpha^2} + G\alpha^{1-1/\mathfrak{p}}K$$

$$= (1-\alpha)^t G + \frac{\rho s^2}{b^2 T^{\frac{2\mathfrak{p}-2}{5\mathfrak{p}-3}}} + \frac{GKb^{\frac{\mathfrak{p}-1}{\mathfrak{p}}}}{T^{\frac{2\mathfrak{p}-2}{5\mathfrak{p}-3}}}$$

$$= (1-\alpha)^t G + \frac{Z}{T^{\frac{2\mathfrak{p}-2}{5\mathfrak{p}-3}}}$$

Again, we have the identity:

$$\log(1-\alpha) \leq -\alpha$$
$$\log\left(G(1-\alpha)^t\right) \leq -t\alpha + \log(G)$$

Therefore, if $t$ satisfies:

$$t \geq \frac{T^{\frac{2\mathfrak{p}}{5\mathfrak{p}-3}}}{b}\left(\frac{2\mathfrak{p}-2}{5\mathfrak{p}-3}\log(T) + \log(G) - \log(Z)\right)$$

we have with probability at least $1-\delta$:

$$\|\hat{\epsilon}_{t+1}\|_\star \leq \frac{2Z}{T^{\frac{2\mathfrak{p}-2}{5\mathfrak{p}-3}}}$$

Next, again from Lemma 1:

$$F(\vec{w}_{t+1}) \leq F(\vec{w}_t) - \eta\|\nabla F(\vec{w}_t)\|_\star + 2\eta\|\hat{\epsilon}_t\|_\star + \frac{L\eta^2}{2}$$

$$\leq F(\vec{w}_t) - \eta\|\vec{m}_t\| + 3\eta\|\hat{\epsilon}_t\|_\star + \frac{L\eta^2}{2}$$

$$\leq F(\vec{w}_t) - \eta\|\vec{m}_t\| + \eta\left(\frac{6Z}{T^{\frac{2\mathfrak{p}-2}{5\mathfrak{p}-3}}} + \frac{Ls}{2T^{\frac{3\mathfrak{p}-1}{5\mathfrak{p}-3}}}\right)$$

Now, analogously to Theorem 3, we argue that if $\|\vec{m}_t\| \geq 2\left(\frac{6Z}{T^{\frac{2\mathfrak{p}-2}{5\mathfrak{p}-3}}} + \frac{Ls}{2T^{\frac{3\mathfrak{p}-1}{5\mathfrak{p}-3}}}\right)$, then $F(\vec{w}_{t+1}) \leq F(\vec{w}_t) - \frac{\eta}{2}\|\vec{m}_t\|$.

The last part of the Theorem follows from an identical case-work argument to that of Theorem 3. $\qquad\square$

## B   A Concentration Bound for Truncated Random Vectors

A key tool in proving our bounds is the following result, which provides a dimension-free concentration bound analogous to the concentration bounds for scalar random variables that can be found in [23, 6]. We present the simpler results for Hilbert space first, and then generalize to the Banach space case.

**Lemma 8.** *Suppose* $X_1, \ldots, X_T$ *are random vectors in a Hilbert space adapted to a filtration* $\mathcal{F}_1, \mathcal{F}_2, \ldots$ *with means* $\mathbb{E}_{t-1}[X_t] = \mu_t$. *Suppose there is some* $\mathfrak{p} \in (1, 2]$ *such that for all $t$ there is some* $G_t < \infty$ *such that* $\mathbb{E}[\|X_t\|^{\mathfrak{p}}] \leq G_t^{\mathfrak{p}}$. *Let* $b_1, \ldots, b_T$ *be fixed constants, with* $B = \max_t b_t$ *such that* $B \leq 1$. *For some* $\tau > 0$, *let* $\hat{X}_t = \frac{X_t}{\|X_t\|}\min(\tau, \|X_t\|)$. *Then with probability at least* $1-\delta$:

$$\left\|\sum_{t=1}^T b_t(\hat{X}_t - \mu_t)\right\| \leq 4B\tau\log(3/\delta) + \sum_{t=1}^T \frac{b_t G_t^{\mathfrak{p}}}{\tau^{\mathfrak{p}-1}} + 2\sqrt{\sum_{t=1}^T b_t^2 G_t^{\mathfrak{p}}\tau^{2-\mathfrak{p}}\max(1, \log(3/\delta))}$$

The proof has two main steps: first, we bound the bias and the variance of the truncated random vectors in terms of the truncation threshsold. Then, we use a dimension-free Feedman-style bound to show that the average of these truncated vectors concentrates about the mean of the original un-truncated vectors.

*Proof.* First, we will bound $\|\mathbb{E}[\hat{X}_t] - \mu_t\|$, following analysis similar to that present in [43]:

$$\|\mathbb{E}[\hat{X}_t] - \mu_t\| = \|\mathbb{E}[\hat{X}_t - X_t]\|$$
$$\leq \mathbb{E}[\|\hat{X}_t - X_t\|]$$
$$\leq \mathbb{E}[\|X_t\|\mathbf{1}[\|X_t\| \geq \tau]]$$
$$\leq \mathbb{E}[\|X_t\|^{\mathfrak{p}}/\tau^{\mathfrak{p}-1}]$$
$$\leq \frac{G_t^{\mathfrak{p}}}{\tau^{\mathfrak{p}-1}}$$

Next, we bound the variance $\mathbb{E}[\|\hat{X}_t - \mathbb{E}[\hat{X}_t]\|^2]$:

$$\mathbb{E}[\|\hat{X}_t - \mathbb{E}[\hat{X}_t]\|^2] \leq \mathbb{E}[\|\hat{X}_t\|^2]$$
$$\leq \mathbb{E}[\|X_t\|^{\mathfrak{p}}\tau^{2-\mathfrak{p}}]$$
$$\leq G_t^{\mathfrak{p}}\tau^{2-\mathfrak{p}}$$

Now, invoke a vector-valued Feedman-inequality (Lemma **??**), observing that $\|\hat{X}_t\| \leq \tau$ always so that $\|b_t(\hat{X}_t - \mathbb{E}[\hat{X}_t])\| \leq 2b_t\tau \leq 2B\tau$ where $B = \max_t b_t$. Thus, with probability $1 - \delta$:

$$\left\|\sum_{t=1}^{T} b_t(\hat{X}_t - \mathbb{E}[\hat{X}_t])\right\| \leq 6B\tau \max(1, \log(3/\delta)) + 3\sqrt{\sum_{t=1}^{T} b_t^2 G_t^{\mathfrak{p}}\tau^{2-\mathfrak{p}} \max(1, \log(3/\delta))}$$

Therefore by triangle inequality:

$$\left\|\sum_{t=1}^{T} b_t(\hat{X}_t - \mu_t)\right\| \leq 6B\tau \max(1, \log(3/\delta)) + \sum_{t=1}^{T} \frac{b_t G_t^{\mathfrak{p}}}{\tau^{\mathfrak{p}-1}} + 3\sqrt{\sum_{t=1}^{T} b_t^2 G_t^{\mathfrak{p}}\tau^{2-\mathfrak{p}} \max(1, \log(3/\delta))}$$

$\square$

Now, we present the generalization to Banach spaces.

**Lemma 9.** *Suppose $X_1, \ldots, X_T$ are random vectors in some Banach space $\mathcal{B}$ satisfying (7) with $p \leq 2$ and $C \geq 1$ adapted to a filtration $F_1, F_2, \ldots$. Let $\mathbb{E}_{t-1}[X_t] = \mu_t$. Suppose there is some $\mathfrak{p} \in (1, 2]$ such that for all $t$ there is some $G_t < \infty$ such that $\mathbb{E}[\|X_t\|^{\mathfrak{p}}] \leq G_t^{\mathfrak{p}}$. Let $b_1, \ldots, b_T$ be fixed constants, with $B = \max_t b_t$ such that $B \leq 1$. For some $\tau > 0$, let $\hat{X}_t = \frac{X_t}{\|X_t\|} \min(\tau, \|X_t\|)$. Then with probability at least $1 - \delta$:*

$$\left\|\sum_{t=1}^{T} b_t(\hat{X}_t - \mu_t)\right\| \leq 10CB\tau \max(1, \log(3/\delta)) + \sum_{t=1}^{T} \frac{b_t G_t^{\mathfrak{p}}}{\tau^{\mathfrak{p}-1}} + 4\left(C\sum_{t=1}^{T} b_t^p G_t^{\mathfrak{p}}\tau^{p-\mathfrak{p}}\right)^{1/p}\sqrt{\max(1, \log(3/\delta))}$$

The proof follows the same structure as the previous result for Hilbert spaces.

*Proof.* Note that the bound

$$\|\mathbb{E}[\hat{X}_t] - \mu_t\| \leq \frac{G_t^{\mathfrak{p}}}{\tau^{\mathfrak{p}-1}}$$

continues to hold as the arguments apply without modification in Banach spaces. For the variance, we have an analogous statement:

$$\mathbb{E}[\|\hat{X}_t - \mathbb{E}[\hat{X}_t]\|^p] \leq \mathbb{E}[\|\hat{X}_t\|^p]$$
$$\leq \mathbb{E}[\|X_t\|^{\mathfrak{p}}\tau^{p-\mathfrak{p}}]$$
$$\leq G_t^{\mathfrak{p}}\tau^{p-\mathfrak{p}}$$

Now, invoke a vector-valued Feedman-inequality (Lemma **??**), observing that $\|\hat{X}_t\| \leq \tau$ always so that $\|b_t(\hat{X}_t - \mathbb{E}[\hat{\mu}_t])\| \leq 2b_t\tau \leq 2B\tau$ where $B = \max_t b_t$. Thus, with probability $1 - \delta$:

$$\left\| \sum_{t=1}^{T} b_t(\hat{X}_t - \mathbb{E}[\hat{X}_t]) \right\| \leq 10CB\tau \max(1, \log(3/\delta)) + 4 \left( C \sum_{t=1}^{k} b_t^p G_t^{\mathfrak{p}} \tau^{p-\mathfrak{p}} \right)^{1/p} \sqrt{\max(1, \log(3/\delta))}$$

Therefore by triangle inequality:

$$\left\| \sum_{t=1}^{T} b_t(\hat{X}_t - \mu_t) \right\| \leq 10CB\tau \max(1, \log(3/\delta)) + \sum_{t=1}^{T} \frac{b_t G_t^{\mathfrak{p}}}{\tau^{\mathfrak{p}-1}} + 4 \left( C \sum_{t=1}^{k} b_t^p G_t^{\mathfrak{p}} \tau^{p-\mathfrak{p}} \right)^{1/p} \sqrt{\max(1, \log(3/\delta))}$$

$\square$

## C  Dimension-Free Martingale concentration from 1-d concentration

In this section, we show how to obtain a dimension-free Freedman-style martingale concentration bound in a Banach space via a reduction to the ordinary 1-dimensional Freedman inequality. The conversion is based on a construction outlined in [8] for converting 1-dimensional online linear optimization algorithms into dimension-free algorithms. It may be of independent interest as simple alternative proof for such concentration results, as well as providing a possibly more user-friendly result in comparison to advanced bounds available in [34, 24, 17]. However, we emphasis that the results in this section are essentially provided only for completeness: they do *not* improve upon previously available bounds.

Suppose $B$ is a real Banach space whose norm $\| \cdot \|$ is Frechet differentiable and satisfies for some $p$ and $C$ for all $x, y$:

$$\|x + y\|^p \leq \|x\|^p + \langle \nabla \|x\|^p, y \rangle + C\|y\|^p \tag{7}$$

Notice that if $B$ is a Hilbert space, we can take $p = 2$ and $C = 1$.

Given any sequence of random vectors $X_1, \ldots, X_T$ in $B$, consider the sequence of real numbers $s_1, \ldots, s_T$ defined recursively by:

1. $s_0 = 0$
2. If $\sum_{i=1}^{t-1} X_i \neq 0$, then we set:

$$s_t = \text{sign}\left( \sum_{i=1}^{t-1} s_i \right) \frac{\langle \nabla \| \sum_{i=1}^{t-1} X_i \|^p, X_t \rangle}{p \| \sum_{i=1}^{t-1} X_i \|^{p-1}},$$

   where $\text{sign}(x) = 1$ if $x \geq 0$, $-1$ if $x < 0$ and $0$ if $x = 0$.
3. If $\sum_{i=1}^{t-1} X_i = 0$, set $s_t = 0$.

The critical property of this sequence is the following, proved in [8] Theorem 5.3. We reproduce the proof for completeness below as we have mildly tightened the constants:

**Lemma 10.** *Suppose $B$, $s_t$ and $X_t$ are as described above and $p \geq 1$. Then $|s_t| \leq \|X_t\|$ for all t, and*

$$\left\| \sum_{t=1}^{T} X_t \right\| \leq \left| \sum_{t=1}^{T} s_t \right| + \left( \max_{t \leq T} \|X_t\|^p + C \sum_{t=1}^{T} \|X_t\|^p \right)^{1/p}$$

*Proof.* For the bound on $s_t$, observe that since $x \mapsto \|x\|$ is by definition 1-Lipschitz, $\|(\nabla \|x\|)\|_\star \leq 1$. Therefore we have

$$|s_t| \leq \frac{|\langle \nabla \| \sum_{i=1}^{t-1} X_i \|^p, X_t \rangle|}{p \| \sum_{i=1}^{t-1} X_i \|^{p-1}}$$

$$\leq \langle \nabla \| \sum_{i=1}^{t-1} X_i \|, X_t \rangle$$

$$\leq \|X_t\|$$

Now, for the second statement we proceed by induction. Clearly the statement holds for $T = 1$. Suppose now that

$$\left| \sum_{t=1}^{T-1} X_t \right| \leq \left\| \sum_{t=1}^{T-1} s_t \right\| + \left( \max_{t \leq T-1} \|X_t\|^p + C \sum_{t=1}^{T-1} \|X_t\|^p \right)^{1/p}$$

Now, first observe that if $\sum_{i=1}^{T-1} X_i = 0$, then $\|\sum_{i=1}^{T} X_i\| = \|X_i\| \leq \max_{t \leq T} \|X_i\|$ and so the statement holds. Let us now assume $\sum_{i=1}^{T-1} X_i \neq 0$ Define $X_{1:t} = \sum_{i=1}^{t} X_i$. Then we have:

$$\left\| \sum_{i=1}^{T} X_i \right\| = (\|X_{1:T-1} + X_T\|^p)^{1/p} \leq (\|X_{1:T-1}\|^p + \langle \nabla \|X_{1:T-1}\|^p, X_T \rangle + C\|X_T\|^p)^{1/p}$$

Now we consider two cases, either $\|X_{1:T-1}\|^p + \langle \nabla \|X_{1:T-1}\|^p, X_T \rangle \leq 0$ or not. If $\|X_{1:T-1}\|^p + \langle \nabla \|X_{1:T-1}\|^p, X_T \rangle \leq 0$, then we have just shown:

$$\left\| \sum_{i=1}^{T} X_i \right\| \leq C^{1/p} \|X_T\| \leq \left| \sum_{i=1}^{T} s_i \right| + \left( \max_{t \leq T} \|X_t\|^p + C \sum_{i=1}^{T} \|X_t\|^p \right)^{1/p}$$

and so we are done.

Instead, let us suppose $\|X_{1:T-1}\|^p + \langle \nabla \|X_{1:T-1}\|^p, X_T \rangle > 0$. This implies $\|X_{1:T-1}\| + \frac{\langle \nabla \|X_{1:T-1}\|^p, X_T \rangle}{p\|X_{1:T-1}\|^{p-1}} \geq 0$ as well. Now, since the function $x \mapsto x^p$ is convex for any positive $x$, we have $x^p + pyx^{p-1} \leq (x + y)^p$ for any $x \geq 0$ and $x + y \geq 0$. Thus:

$$\|X_{1:T-1}\|^p + \langle \nabla \|X_{1:T-1}\|^p, X_T \rangle \leq \left( \|X_{1:T-1}\| + \frac{\langle \nabla \|X_{1:T-1}\|^p, X_T \rangle}{p\|X_{1:T-1}\|^{p-1}} \right)^p$$

Putting this together with our induction hypothesis:

$$\left\| \sum_{i=1}^{T} X_i \right\| \leq \left( \left( \|X_{1:T-1}\| + \frac{\langle \nabla \|X_{1:T-1}\|^p, X_T \rangle}{p\|X_{1:T-1}\|^{p-1}} \right)^p + C\|X_T\|^p \right)^{1/p}$$

$$\leq \left( \left( \left| \sum_{i=1}^{T-1} s_i \right| + \left( \max_{i \leq T-1} \|X_i\|^p + C \sum_{i=1}^{T-1} \|X_i\|^p \right)^{1/p} + \frac{\langle \nabla \|X_{1:T-1}\|^p, X_T \rangle}{p\|X_{1:T-1}\|^{p-1}} \right)^p + C\|X_T\|^p \right)^{1/p}$$

$$\leq \left[ \left( \left( \left| \sum_{i=1}^{T-1} s_i \right| + \frac{\langle \nabla \|X_{1:T-1}\|^p, X_T \rangle}{p\|X_{1:T-1}\|^{p-1}} \right) + \left( \max_{i \leq T-1} \|X_i\|^p + C \sum_{i=1}^{T-1} \|X_i\|^p \right)^{1/p} \right)^p + C\|X_T\|^p \right]^{1/p}$$

Now, following [8], we observe that for any positive $a$, $b$ and $c$, $(a+b)^p - b^p \leq (a+b+c)^p - (b+c)^p$. Thus, setting $a = \left| \sum_{i=1}^{T-1} s_i \right| + \frac{\langle \nabla \|X_{1:T-1}\|^p, X_T \rangle}{p\|X_{1:T-1}\|^{p-1}} \right|$, $b = \left( \max_{i \leq T-1} \|X_i\|^p + C \sum_{i=1}^{T-1} \|X_i\|^p \right)^{1/p}$ and $b + c = \left( \max_{i \leq T} \|X_i\|^p + C \sum_{i=1}^{T} \|X_i\|^p \right)^{1/p}$, we obtain:

$$\left( \left| \sum_{i=1}^{T-1} s_i \right| + \frac{\langle \nabla \|X_{1:T-1}\|^p, X_T \rangle}{p\|X_{1:T-1}\|^{p-1}} \right| + \left( \max_{i \leq T-1} \|X_i\|^p + C \sum_{i=1}^{T-1} \|X_i\|^p \right)^{1/p} \right)^p \leq (a+b+c)^p + b^p - (b+c)^p$$

$$= (a+b+c)^p - C\|X_T\|^p$$

Plugging this identity back into our previous bound on $\left\| \sum_{i=1}^{T} X_i \right\|$, we have:

$$\left\| \sum_{i=1}^{T} X_i \right\| \leq \left| \sum_{i=1}^{T-1} s_i \right| + \frac{\langle \nabla \|X_{1:T-1}\|^p, X_T \rangle}{p\|X_{1:T-1}\|^{p-1}} \right| + \left( \max_{i \leq T} \|X_i\|^p + C \sum_{i=1}^{T} \|X_i\|^p \right)^{1/p}$$

$$= \left| \sum_{i=1}^{T-1} s_i \right| + \mathrm{sign} \left( \sum_{i=1}^{t-1} s_i \right) s_T \right| + \left( \max_{i \leq T} \|X_i\|^p + C \sum_{i=1}^{T} \|X_i\|^p \right)^{1/p}$$

$$= \left| \sum_{i=1}^{T} s_i \right| + \left( \max_{i \leq T} \|X_i\|^p + C \sum_{i=1}^{T} \|X_i\|^p \right)^{1/p}$$

$\square$

### C.1 Warm-up: Hilbert space

Now, we show how to use Lemma 10 in conjuction with the standard 1-dimensional Freedman inequality to generate concentration bounds in Hilbert spaces. First, we state a (mildly weaker) version of the standard 1-dimensional inequality.

**Lemma 11.** *[Freedman's inequality] Suppose $D_1, D_2, \ldots, D_T$ is a martingale difference sequence adapted to a filtration $F_1, F_2, \ldots$ such that $D_i \leq R$ almost surely for all $i$. Let $\mathbb{E}_i$ indicate expectation conditioned on $F_i$. Suppose further that for all $t$ with probability 1,*

$$\sigma_t^2 \geq \mathbb{E}_{t-1}[D_t^2]$$

*Then with probability at least $1 - \delta$, for all $k$ we have*

$$\sum_{t=1}^{k} D_t \leq \frac{2R \log(1/\delta)}{3} + \sqrt{2 \sum_{t=1}^{k} \sigma_t^2 \log(1/\delta)}$$

*Proof.* From the standard inequality (see e.g. [33] Theorem 1.1), we have

$$P\left[\exists k \; : \; \sum_{t=1}^{k} D_t \geq \epsilon\right] \leq \exp\left(-\frac{\epsilon^2/2}{\sum_{t=1}^{k} \sigma_t^2 + R\epsilon/3}\right)$$

Now, set the RHS equal to $\delta$ to obtain:

$$\frac{\epsilon^2/2}{\sum_{t=1}^{k} \sigma_t^2 + R\epsilon/3} = \log(1/\delta)$$

The result now follows by using the quadratic formula to bound $\epsilon$. $\square$

Now, we are in a position to describe our extension of Freedman's inequality to Hilbert spaces (which satisfy the hypotheses of Lemma 10 with $p = 2$ and $C = 1$):

**Lemma 12.** *Suppose $X_1, \ldots, X_T$ is a martingale difference sequence in a Hilbert space such that $\|X_t\| \leq R$ almost surely for some constant $R$. Further, assume $\mathbb{E}_{t-1}[\|X_t\|^2] \leq \sigma_t^2$ with probability 1 for some constants $\sigma_t$. Then with probability at least $1 - 3\delta$, for all $k$ we have:*

$$\left\|\sum_{t=1}^{k} X_t\right\| \leq 3R \max(1, \log(1/\delta))$$

$$+ 3\sqrt{\sum_{t=1}^{k} \sigma_t^2 \max(1, \log(1/\delta))}$$

*Proof.* Define $s_t$ as in Lemma 10. Then with probability 1 we have for all $k$:

$$\left\|\sum_{t=1}^{k} X_t\right\| \leq \left|\sum_{t=1}^{k} s_t\right| + \sqrt{R^2 + \sum_{t=1}^{K} \|X_t\|^2}$$

Further, notice that $s_t$ is itself a martingale difference sequence, and satisfies $|s_t| \leq \|X_t\| \leq R$. Therefore by Lemma 11, with probability at least $1 - 2\delta$

$$\left|\sum_{t=1}^{k} s_t\right| \leq \frac{2R \log(1/\delta)}{3} + \sqrt{2 \sum_{t=1}^{k} \sigma_t^2 \log(1/\delta)}$$

Now, for the second term, set $Z_t = \|X_t\|^2 - \mathbb{E}_{t-1}[\|X_t\|^2]$. Then $Z_t$ is also a martingale difference sequence. Since $\|X_t\|^2 \leq R^2$ with probability 1, $Z_t \leq R^2$ with probability 1. Further,

$$\mathbb{E}_{t-1}[\|Z_t\|^2] \leq \mathbb{E}_{t-1}[\|X_t\|^4] \leq \mathbb{E}_{t-1}[\|X_t\|^2]R^2 \leq \sigma_t^2 R^2$$

Thus, again by Lemma 11, with probability at least $1 - \delta$:

$$\sum_{t=1}^{k} Z_t \leq \frac{2R^2 \log(1/\delta)}{3} + \sqrt{2 \sum_{t=1}^{T} R^2 \sigma_t^2 \log(1/\delta)}$$

which implies:

$$\sum_{t=1}^{k} \|X_t\|^2 \leq \sum_{t=1}^{k} \sigma_t^2 + \frac{2R^2 \log(1/\delta)}{3} + \sqrt{2 \sum_{t=1}^{k} R^2 \sigma_t^2 \log(1/\delta)}$$

Applying Young inequality:

$$\leq \sum_{t=1}^{k} \sigma_t^2 + \frac{2R^2 \log(1/\delta)}{3} + \sqrt{R^4 \log^2(1/\delta) + \left(\sum_{t=1}^{k} \sigma_t^2\right)^2}$$

$$\leq 2 \sum_{t=1}^{k} \sigma_t^2 + \frac{5R^2 \log(1/\delta)}{3}$$

Putting everything together, with probability at least $1 - 3\delta$:

$$\left\| \sum_{t=1}^{k} X_t \right\| \leq \left| \sum_{t=1}^{k} s_t \right| + \sqrt{R^2 + \sum_{t=1}^{K} \|X_t\|^2}$$

$$\leq \frac{2R(\log(1/\delta) + 3/2)}{3} + \sqrt{2 \sum_{t=1}^{k} \sigma_t^2 \log(1/\delta)} + \sqrt{2 \sum_{t=1}^{k} \sigma_t^2 + \frac{5R^2 \log(1/\delta)}{3}}$$

$$\leq 3R \max(1, \log(1/\delta)) + 3 \sqrt{\sum_{t=1}^{k} \sigma_t^2 \max(1, \log(1/\delta))}$$

$\square$

## C.2 Extension to Banach space

Having provided the concentration results in the more familiar Hilbert space setting, now we move to Banach spaces. We will also need the following useful observation:

**Lemma 13.** *For any $0 < p \leq q$ and any positive $x_1, \ldots, x_T$,*

$$\left( \sum_{t=1}^{T} x_t^q \right)^{1/q} \leq \left( \sum_{t=1}^{T} x_t^p \right)^{1/p}$$

*Proof.* We differentiate the expression $\left( \sum_{t=1}^{T} x_t^p \right)^{1/p}$ with respect to $p$ and show that the derivative is always negative, which suffices to prove the Lemma.

$$\frac{d}{dp}\left(\sum_{t=1}^{T}x_t^p\right)^{1/p} = \frac{d}{dp}\exp\left\{\frac{1}{p}\log\left[\sum_{t=1}^{T}\exp\left(p\log(x_t)\right)\right]\right\}$$

$$= \left(\sum_{t=1}^{T}x_t^p\right)^{1/p}\left(\frac{-\log\left[\sum_{t=1}^{T}\exp\left(p\log(x_t)\right)\right]}{p^2} + \frac{\sum_{t=1}^{T}\exp\left(p\log(x_t)\right)\log(x_t)}{p\sum_{t=1}^{T}\exp\left(p\log(x_t)\right)}\right)$$

$$= \left(\sum_{t=1}^{T}x_t^p\right)^{1/p}\left(\frac{-\log\left[\sum_{t=1}^{T}x_t^p\right]}{p^2} + \frac{\sum_{t=1}^{T}x_t^p\log(x_t)}{p\sum_{t=1}^{T}x_t^p}\right)$$

$$= \frac{\left(\sum_{t=1}^{T}x_t^p\right)^{1/p}}{p^2}\left(-\log\left[\sum_{t=1}^{T}x_t^p\right] + \frac{\sum_{t=1}^{T}x_t^p\log(x_t^p)}{\sum_{t=1}^{T}x_t^p}\right)$$

$$= \frac{\left(\sum_{t=1}^{T}x_t^p\right)^{1/p-1}}{p^2}\left(\sum_{t=1}^{T}x_t^p\log(x_t^p) - \log\left[\sum_{t=1}^{T}x_t^p\right]\left(\sum_{t=1}^{T}x_t^p\right)\right)$$

since $\log$ is an increasing function and $x_t > 0$:

$$\leq \frac{\left(\sum_{t=1}^{T}x_t^p\right)^{1/p-1}}{p^2}\left(\sum_{t=1}^{T}x_t^p\log\left[\sum_{t=1}^{T}x_t^p\right] - \log\left[\sum_{t=1}^{T}x_t^p\right]\left(\sum_{t=1}^{T}x_t^p\right)\right)$$

$$= 0$$

$\square$

**Lemma 14.** *Suppose $X_1, \ldots, X_T$ is a martingale difference sequence in a Banach space satisfying (7) for $p \in (1,2]$ such that $\|X_t\| \leq R$ almost surely for some constant $R$. Further, assume $\mathbb{E}_{t-1}[\|X_t\|^2] \leq \sigma_t^2$ with probability 1 for some constants $\sigma_t$. Then with probability at least $1 - 3\delta$, for all $k$ we have:*

$$\left\|\sum_{t=1}^{k}X_t\right\| \leq \frac{2R(\log(1/\delta)+3/2)}{3} + \sqrt{2\sum_{t=1}^{k}\sigma_t^2\log(1/\delta)} + \left(2C\sum_{t=1}^{k}\sigma_t^p + \frac{7CR^p\log(1/\delta)}{3}\right)^{1/p}$$

*Additionally, if $C \geq 1$ and $p \leq 2$:*

$$5CR\max(1,\log(3/\delta)) + 4\left(C\sum_{t=1}^{k}\sigma_t^p\right)^{1/p}\sqrt{\max(1,\log(3/\delta))}$$

*Proof.* First, note that by Jensen, we have $\mathbb{E}_{t-1}[\|X_t\|^p] \leq \mathbb{E}_{t-1}[\|X_t\|^2]^{p/2} \leq \sigma_t^p$.

Define $s_t$ as in Lemma 10. Then with probability 1 we have for all $k$:

$$\left\|\sum_{t=1}^{k}X_t\right\| \leq \left|\sum_{t=1}^{k}s_t\right| + \left(R^p + C\sum_{t=1}^{K}\|X_t\|^p\right)^{1/p}$$

Further, notice that $s_t$ is itself a martingale difference sequence, and satisfies $|s_t| \leq \|X_t\| \leq R$. Therefore by Lemma 11, with probability at least $1 - 2\delta$

$$\left|\sum_{t=1}^{k}s_t\right| \leq \frac{2R\log(1/\delta)}{3} + \sqrt{2\sum_{t=1}^{k}\sigma_t^2\log(1/\delta)}$$

Now, for the second term, set $Z_t = \|X_t\|^p - \mathbb{E}_{t-1}[\|X_t\|^p]$. Then $Z_t$ is also a martingale difference sequence. Since $\|X_t\|^p \leq R^p$ with probability 1, $Z_t \leq 2R^p$ with probability 1. Further,

$$\mathbb{E}_{t-1}[\|Z_t\|^2] \leq \mathbb{E}_{t-1}[\|X_t\|^{2p}] \leq \mathbb{E}_{t-1}[\|X_t\|^p]R^p \leq \sigma_t^p R^p$$

Thus, again by Lemma 11, with probability at least $1 - \delta$:

$$\sum_{t=1}^{k} Z_t \leq \frac{4R^p \log(1/\delta)}{3} + \sqrt{2 \sum_{t=1}^{T} R^p \sigma_t^p \log(1/\delta)}$$

$$\sum_{t=1}^{k} \|X_t\|^p \leq \sum_{t=1}^{k} \sigma_t^p + \frac{4R^p \log(1/\delta)}{3} + \sqrt{2 \sum_{t=1}^{k} R^p \sigma_t^p \log(1/\delta)}$$

Applying Young inequality:

$$\leq \sum_{t=1}^{k} \sigma_t^p + \frac{4R^p \log(1/\delta)}{3} + \sqrt{R^{2p} \log^2(1/\delta) + \left(\sum_{t=1}^{k} \sigma_t^p\right)^2}$$

$$\leq 2 \sum_{t=1}^{k} \sigma_t^p + \frac{7R^p \log(1/\delta)}{3}$$

Putting everything together, with probability at least $1 - 3\delta$:

$$\left\| \sum_{t=1}^{k} X_t \right\| \leq \left| \sum_{t=1}^{k} s_t \right| + \left( R^p + C \sum_{t=1}^{K} \|X_t\|^p \right)^{1/p}$$

$$\leq \frac{2R(\log(1/\delta) + 3/2)}{3} + \sqrt{2 \sum_{t=1}^{k} \sigma_t^2 \log(1/\delta)} + \left( 2C \sum_{t=1}^{k} \sigma_t^p + \frac{7CR^p \log(1/\delta)}{3} \right)^{1/p}$$

Now, if $C \geq 1$, we can apply Lemma 13 and over approximate to obtain with probability $1 - 3\delta$:

$$\left\| \sum_{t=1}^{k} X_t \right\| \leq \frac{2R(\log(1/\delta) + 3/2)}{3} + \left( \sum_{t=1}^{k} \sigma_t^p \right)^{1/p} \sqrt{2 \log(1/\delta)} + \left( 2C \sum_{t=1}^{k} \sigma_t^p \right)^{1/p} + \left( \frac{7CR^p \log(1/\delta)}{3} \right)^{1/p}$$

$$\leq 5CR \max(1, \log(1/\delta)) + 4 \left( C \sum_{t=1}^{k} \sigma_t^p \right)^{1/p} \sqrt{\max(1, \log(1/\delta))}$$

$\square$

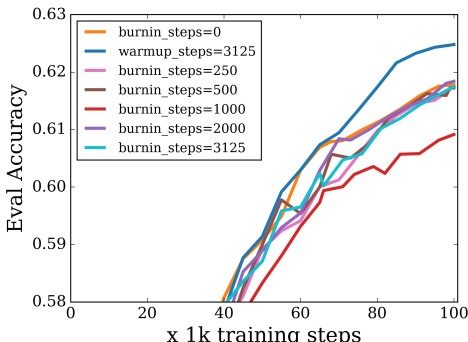 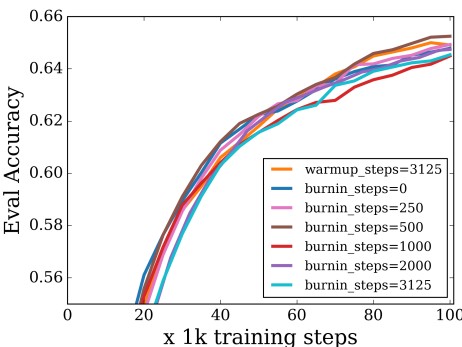

Figure 2: Comparison between warm-up and burn-in strategies on Masked language modeling BERT-Base pre-training task. Plot (a) shows the eval accuracy when trained with Algorithm 1 and plot (b) does the same when trained with LAMB.

## D    Experiments

In this section, we run additional experiments on BERT-Base pre-training task. BERT-Base is a smaller version of BERT-Large model we experimented in Section 4. Again, all our experiments were conducted using the Tensorflow framework on TPUv3 architecture. When using Algorithm 1, we train all the models shown using the base learning rate $\eta_0$ of 0.5 and batch size of 512. We came up with this learning rate by performing grid search $\eta_0 \in [1.0, 0.5, 0.1, 0.05, 0.01, 0.005, 0.001, 0.0005, 0.0001]$ and choosing the one which attains the best eval accuracy. Again, shown in Fig 2, our warmup baseline employs the standard practice of linear warm-up for 3125 steps and polynomical decay of $\eta_t = \eta_0 * (1 - \frac{t}{T})$ for the rest of the steps. For a fair comparison with the baseline, we start the polynomial decay after 3125 steps regardless of how many steps were used for burn-in. We set the learning rate to the constant base learning rate value of 0.5 for for steps in between burn-in and polynomial decay. For the models trained using burn-in, instead of linear warm-up, we set the base learning rate $\eta_0$ to zero for the amount of steps we run the burn-in for. As shown in Fig 2, we perform a grid search across $[0, 250, 500, 1000, 2000, 3125]$ steps for the initial burn-in period and compare their results. When training the smaller BERT-Base model we observe that burn-in is less crucial compared to when training BERT-Large i.e. the model converges regardless of the amount of burn-in employed. Moreover, we don't see any convergence issues when we don't employ even any warm-up, this suggests that warm-up is also much less crucial (similar to burn-in) on the simpler BERT-Base problem.

## E    Smoothness Lemmas in Banach Spaces

In this section we provide a few technical lemmas relating to smoothness in Banach spaces. Most of these are fairly straightforward generalizations of standard results in Hilbert spaces.

We begin with a notations and definitions. Recall that a *Banach space* over the reals is a real vector space $\mathcal{B}$ together with a norm $\|\cdot\| : \mathcal{B} \to \mathbb{R}$ such that $\mathcal{B}$ is complete with respect to the topology induced by the norm. The dual space $\mathcal{B}^\star$ is the space of bounded linear functions $\mathcal{B} \to \mathbb{R}$. When $\mathcal{B}$ is a Hilbert space, there is a natural isomorphism between $i : \mathcal{B}^\star \to \mathcal{B}$ such that for any $v \in \mathcal{B}^\star$ and $w \in \mathcal{B}$, $v(w) = \langle i(v), w \rangle$. In order to preserve notational familiarity, even when $\mathcal{B}$ is *not* a Hilbert space, we therefore adopt the notation $\langle v, w \rangle$ to refer to $v(w)$, the application of the linear function $v \in \mathcal{B}^\star$ to the vector $w \in \mathcal{B}$. Note that in this way the ordering of the arguments to $\langle \cdot, \cdot \rangle$ is important: $\langle w, v \rangle$ is meaningless. The dual space $\mathcal{B}^\star$ is equipped with the *dual norm* $\|v\|_\star = \sup_{\|x\| \leq 1} \langle v, x \rangle$, and is itself a Banach space.

A function $F : \mathcal{B} \to \mathbb{R}$ is Frechet differentiable if for all $w \in \mathcal{B}$ there is a bounded linear operator $v \in \mathcal{B}^\star$ such that:

$$\lim_{\delta \to 0} \frac{|F(w + \delta) - F(w) - \langle v, \delta \rangle|}{\|\delta\|} = 0$$

the operator $v$ is denoted $\nabla F(w)$. Note that if $F$ is Frechet differentiable, it also satisfies the familiar directional derivative identity: if $f(t) = F(w + t\delta)$, then

$$f'(t) = \langle \nabla F(w + t\delta), \delta \rangle$$

The second derivative or Hessian of $F$ at a point $w$ is a linear map $\nabla^2 F(w) : \mathcal{B} \to \mathcal{B}^\star$ satisfying:

$$\lim_{\delta \to 0} \frac{\|\nabla F(w + \delta) - \nabla F(w) - \nabla^2 F(w)\delta\|_\star}{\|\delta\|} = 0$$

In this paper we exclusively consider spaces such that $\mathcal{B}^\star$ satisfies:

$$\|x + y\|_\star^p \leq \|x\|_\star^p + \langle \nabla \|x\|_\star^p, y \rangle + C\|y\|_\star^p \tag{8}$$

for some $p$ and $C$ for all $x, y \in \mathcal{B}^\star$. We will also assume that $C \geq 1$.

**Proposition 15.** *Suppose that $F : \mathcal{B} \to \mathbb{R}$ is Frechet differentiable and the gradient satisfies $\|\nabla F(x) - \nabla F(y)\|_\star \leq L\|x - y\|$ for all $x, y \in \mathcal{B}$ (i.e. $F$ is L-smooth). Then for all $x, y \in \mathcal{B}$:*

$$F(x) \leq F(y) + \langle \nabla F(y), x - y \rangle + \frac{L}{2}\|x - y\|^2$$

*Proof.* Consider the function $g : \mathbb{R} \to \mathbb{R}$ given by $g(t) = F(y + t(x - y))$. Since $F$ is Frechet differentiable, $g$ is differentiable and $g'(t) = \langle \nabla F(y + t(x - y)), (x - y) \rangle$. Therefore by the fundamental theorem of calculus:

$$g(1) = g(0) + \int_0^1 g'(t)dt$$

$$= g(0) + g'(0) + \int_0^1 (g'(t) - g'(0))dt$$

$$\leq g(0) + \langle \nabla F(y), x - y \rangle + \int_0^1 |g'(t) - g'(0)|dt$$

$$\leq g(0) + \langle \nabla F(y), x - y \rangle + \int_0^1 |\langle \nabla F(y + t(x - y)) - \nabla F(y), x - y \rangle|dt$$

$$\leq g(0) + \langle \nabla F(y), x - y \rangle + \int_0^1 tL\|x - y\|^2 dt$$

$$= g(0) + \langle \nabla F(y), x - y \rangle + \frac{L}{2}\|x - y\|^2|$$

Now plug in the definition of $g(1)$ and $g(0)$ to conclude the argument. $\square$

**Proposition 16.** *Suppose that $F : \mathcal{B} \to \mathbb{R}$ is twice Frechet differentiable and the Hessian satisfies $\|(\nabla^2 F(x) - \nabla^2 F(y))v\|_\star \leq \rho\|v\|\|x - y\|$ for all $x, y, v \in \mathcal{B}$ (i.e. $F$ is $\rho$-second-order smooth). Then for all $x, y \in \mathcal{B}$:*

$$F(x) \leq F(y) + \langle \nabla F(y), x - y \rangle + \frac{\langle \nabla^2 F(y)(x - y), x - y \rangle}{2} + \frac{\rho}{6}\|x - y\|^3$$

*Proof.* Consider the function $g : \mathbb{R} \to \mathbb{R}$ given by $g(t) = F(y + t(x - y))$. Since $F$ is twice Frechet differentiable, $g$ is twice differentiable and $g'(t) = \langle \nabla F(y + t(x - y)), x - y \rangle$ and $g''(t) = \langle \nabla^2 F(y + t(x - y))(x - y), x - y \rangle$. Therefore by the fundamental theorem of calculus:

$$g(1) = g(0) + \int_0^1 g'(t)dt$$

$$= g(0) + \int_0^1 \left( g'(0) + \int_0^t g''(k)dk \right) dt$$

$$= g(0) + g'(0) + \int_0^1 \int_0^t g''(k)dkdt$$

$$= g(0) + g'(0) + \int_0^1 \int_0^t g''(0)dkdt + \int_0^1 \int_0^t (g''(k) - g''(0))dkdt$$

$$= g(0) + g'(0) + \frac{g''(0)}{2} + \int_0^1 \int_0^t (g''(k) - g''(0))dkdt$$

$$= g(0) + g'(0) + \frac{g''(0)}{2} + \int_0^1 \int_0^t \langle (\nabla^2 F(y + k(x - y)) - \nabla^2 F(y))(x - y), x - y \rangle dkdt$$

$$\leq g(0) + g'(0) + \frac{g''(0)}{2} + \int_0^1 \int_0^t \rho\|x - y\|^3 kdkdt$$

$$\leq g(0) + g'(0) + \frac{g''(0)}{2} + \frac{\rho\|x - y\|^3}{6}$$

Now plug in the definition of $g(1)$ and $g(0)$ to conclude the argument. $\square$

**Proposition 17.** *Suppose that $F : \mathcal{B} \to \mathbb{R}$ is twice Frechet differentiable and the Hessian satisfies $\|(\nabla^2 F(x) - \nabla^2 F(y))v\|_\star \leq \rho\|v\|\|x - y\|$ for all $x, y, v \in \mathcal{B}$ (i.e. $F$ is $\rho$-second-order smooth). Then for all $x, y \in \mathcal{B}$:*

$$\|\nabla F(y) - F(x) - \nabla^2 F(x)(y - x)\|_\star \leq \frac{\rho}{2}\|x - y\|^2$$

*Proof.* For any fixed $z \in \mathcal{B}$, define $g_z(t) = \langle \nabla F(x + t(y - x)), z \rangle$. Then we have $g'_z(t) = \langle \nabla^2 F(x + t(y - x))(y - x), z \rangle$. Then, by the fundamental theorem of calculus:

$$g_z(1) = g_z(0) + \int_0^1 g'_z(t) dt$$

$$\leq g_z(0) + g'_z(0) + \int_0^1 |g'_z(t) - g'_z(0)| dt$$

$$\leq g_z(0) + g'_z(0) + \int_0^1 \rho t \|x - y\|^2 \|z\| dt$$

$$\leq g_z(0) + g'_z(0) + \frac{\rho \|x - y\|^2 \|z\|}{2}$$

rearrange and apply the definition of $g_z$:

$$\langle \nabla F(y) - \nabla F(x) - \nabla^2 F(x)(y - x), z \rangle \leq \frac{\rho \|x - y\|^2 \|z\|}{2}$$

Since $z$ was arbitrary, by definition of dual norm we have

$$\|\nabla F(y) - \nabla F(x) - \nabla^2 F(x)(y - x)\|_\star \leq \frac{\rho}{2} \|x - y\|^2$$

$\square$

The following important Lemma generalizes Lemma 2 of [9] to more general norms, and improves constants.

**Lemma 1.** *Suppose $F : \mathcal{B} \to \mathbb{R}$ be a Frechet-differentiable L-Smooth function from a Banach space $\mathcal{B}$ to the reals. Let $w \in B$. Let $g^\star \in \mathcal{B}^\star$ and let $g \in \mathcal{B}$ be a unit-vector satisfying $\langle g^\star, g \rangle = \|g\|_\star$. Define $w' = w - \eta g$. Define $\epsilon = g^\star - \nabla F(w)$. Then:*

$$F(w') \leq F(w) - \eta \|\nabla F(w)\|_\star + 2\eta \|\epsilon\|_\star + \frac{L\eta^2}{2}$$

*Proof.* First, by smoothness (Proposition 15) we have

$$F(w') \leq F(w) + \langle \nabla F(w), w' - w \rangle + \frac{L}{2} \|w' - w\|^2$$

Next, since $\|g\| = 1$,

$$F(w') \leq F(w) - \eta \langle \nabla F(w), g \rangle + \frac{L\eta^2}{2}$$

$$= F(w) - \eta \langle g^\star - \epsilon, g \rangle + \frac{L\eta^2}{2}$$

$$= F(w) - \eta (\|g^\star\|_\star - \langle \epsilon, g \rangle) + \frac{L\eta^2}{2}$$

$$\leq F(w) - \eta \|\nabla F(w) + \epsilon\|_\star + \eta \|\epsilon\|_\star \|g\| + \frac{L\eta^2}{2}$$

$$\leq F(w) - \eta \|\nabla F(w)\|_\star + 2\eta \|\epsilon\|_\star + \frac{L\eta^2}{2}$$

$\square$