# OpenReview forum: "High-probability Bounds for Non-Convex Stochastic Optimization with Heavy Tails"
_NeurIPS.cc/2021/Conference — NeurIPS 2021 Oral_

### Official Review · Reviewer_5L1D · 2021-07-14

**Rating:** 7
**Confidence:** 3

**Summary:**

The paper studies non-convex stochastic optimization using normalized SGD with clipping and momentum. They obtain convergence of the average to critical points in high-probability with best-known rates for smooth losses when the gradients have heavy tails, that is, with finite p-th moments for some p\in(1,2], and convergence of the last iterate is also obtained. The paper also considers normalized implicit gradient with transport with clipping in the case of second-order smooth losses, which is new and they obtain convergence results for the average as well as the last iterate.

**Limitations And Societal Impact:**

I could not find out where the authors mention limitations and societal impact in their paper. One limitation I can see is that in the main results, they seem to be asymptotic instead of non-asymptotic.

**Main Review:**

The paper is original and the main results are clear. It is known that the analysis for gradients with heavy tails can be more challenging, and it is nice that the authors can carry out the analysis when the gradient has finite p-th moments for p possibly smaller than 2. The normalized implicit gradient with transport with clipping for second-oder smooth losses is also new.

When I'm reading your proof of Theorem 2 and Theorem 3 for example, it seems you are able to keep track of the constants, and I'm wondering if your results of Theorem 2 and Theorem 3 can be made non-asymptotic with explicit constants instead of big O? That can strengthen the contribution of the paper.

Also, the paper seems to be written in a rush, and there are numerous typos and places with confusions, and I suggest the authors check the paper carefully and paper would be in a better shape if the typos and minor mistakes are corrected. For example,

(1) Page 1. Line 24. for some \sigma. I don't understand this. What's the relation between \sigma and G?

(2) Page 2. Line 45. What do you mean by convergence rate itself is a random variable? For SGD, even though you have randomness, using criteria like Wasserstein metric or others, there are many results in the literature with deterministic convergence rates.

(3) Page 3. In the last paragraph of Section 1.1. What is \nabla F(w,z)? Should it be \nabla F(w) or?

(4) Page 3. In the first paragraph in Section 2, Banac should be Banach.

(5) In Theorem 2, is G the same as the G on page 1?

(6)  Page 4. In the paragraph before Proof of Theorem 2, is burnin-in period a typo?

(7) Page 6. In the second display of Proof of Theorem 3, there is an extra ) for \log(G(1-\apha)^{t}))

(8) You wrote in the proof of Theorem 3, "observe that since F is G-Lipschitz". Where did you assume F is G-Lipschitz?

(9) In the statement of Theorem 5, you wrote "for all x for some G". Should it be "for all w for some G" instead?

(10) In references, some words should be capitalized. For example, sgd should be SGD in [9].

**Time Spent Reviewing:**

2.5

---

> ### Author Response · Authors · 2021-08-10
> **response to reviewer 5L1D**
>
> Thanks very much for your detailed review, and we are gratified to hear your positive comments originality. We apologize for the presentation issues - please rest assured that we will go over the paper very carefully to correct the issues you identified as well as any others before the next draft. Please find below specific responses to your comments:
>
> First, with regards to explicit constants: Yes, we absolutely can do this! As you observed, our proofs do keep track of all the constants, so that we did in fact prove non-asymptotic results. We chose not to detail the constants in the theorem statements of our current draft because we felt they might be complicated-looking enough to distract from the main message. However, we agree that having the explicit constants is valuable, so we will include versions of the results that include all the constants in the revision.
>
> In regards to your line comments:
> (1) It should say G, not $\sigma$.
> (2) We mean to say that the quantities measured in convergence analyses, like $\|\nabla F(\hat w)\|$ or $F(\hat w)-F(w_\star)$ are actually random. Most results in the literature consider bounds on the expectation of these random quantities. Our use of the phrase “convergence rate” was unclear here - thank you for point it out and we will rectify the problem in the revision.
>
> (3) In fact, there should be no second term in that expression (it should just be $\|\nabla f(w, z)\|^p$).
>
> (5) Yes, although the prose on page one is very high-level and only considers p=2.
>
> (6) yes it should just be burn-in period.
>
> (7) Thanks, we will fix it.
>
> (8) We are assuming that $\mathbb{E}[\|\nabla f(w, z)\|^p]\le G^p$. By Jensen inequality, this implies $G^p\ge \|\mathbb{E}[\nabla f(w, z)]\|^p=\|\nabla F(w)\|^p$, so that $F$ is $G$-Lipschitz.
>
> (9) Yes, it should say $w$ instead of $x$.
>
> (10). Thanks we will fix this as well.

---

> > ### Comment · Reviewer_5L1D · 2021-08-23
> > **response to authors**
> >
> > I'm satisfied with this paper. I increased my score. Please make sure you include explicit constants somewhere either in the main body or appendix of the paper, and fix all the typos!

---

### Official Review · Reviewer_KPvy · 2021-07-15

**Rating:** 8
**Confidence:** 5

**Summary:**

The paper contains novel strong theoretical results on finding stationary points for smooth non-convex stochastic problems with heavy-tailed noise. To achieve such results, the authors propose a new method based on normalized Momentum-SGD with clipping of stochastic gradients. The key part of the analysis is in an accurate application of Freedman's inequality for bounded martingale difference sequences with bounded conditional variances. The derived rates have $\log(1/\delta)$ dependence on the probability margin/confidence level $\delta$ that is a desirable property for high-probability results; the dependence on the number of iterations $T$ coincide in the obtained rates is the same as in the state-of-the-art in-expectation convergence results under the same assumptions. Moreover, the paper contains an analysis under second-order smoothness and results for last-iterate convergence that partially explain the success of the "warm-up" stepsize schedules.

Although the paper contains a number of mathematical inaccuracies and typos, I believe that all of them can be easily fixed (see the details below). If the authors address all the comments properly, I will increase my score to 8.

**Limitations And Societal Impact:**

The authors adequately addressed the limitations and potential negative societal impact of their work.

**Main Review:**

## Strengths

1. **The paper focuses on a poorly investigated but very important setup.** In contrast, to almost all previous high-probability results for smooth non-convex stochastic problems, the results derived in this paper hold even in the case when the variance of the stochastic gradient is unbounded, i.e., the authors assume that $\mathbb{E}\|\nabla f(w,z)\|_*^{\mathfrak{p}} \leq G^{\mathfrak{p}}$ for some ${\mathfrak{p}} \in (1,2]$. When ${\mathfrak{p}} < 2$ the noise in the stochastic gradient is heavy-tailed and this setup was never considered in the literature on high-probability convergence. However, obtaining tight high-probability guarantees in the heavy-tailed regime is an important problem. Moreover, for Theorems 5 and 6 there are no known analogs for the convergence in expectation.

2. **Tightness of the obtained results.** The result of Theorem 2 establishes $O\left(\frac{\log(T/\delta)}{T^{\frac{\mathfrak{p}-1}{3\mathfrak{p}-2}}}\right)$ bound for the average of gradients norms after $T$ iterations with probability at least $1-\delta$. This result is obtained under smoothenss of the objective $F$ and $\mathbb{E}\|\nabla f(w,z)\|_*^{\mathfrak{p}} \leq G^{\mathfrak{p}}$ for some ${\mathfrak{p}} \in (1,2]$. In this settings, thereare no known high-probability guarantees in the literature. Moreover, the obtained result has good dependence on $\delta$, and in terms of the dependence on $T$ it matches the "in-expectation" result from [42] that was proven to be optimal (see [42]).

3. **Last-iterate guarantees and the connections with warm-up stepsize schedules.** The authors propose a simple but elegant way to obtain last-iterate convergence guarantees and formalize it as Theorem 3: the function value in the last point produced by the algorithm is not bigger than the function value at the point where the gradient is small enough with high probability. In contrast, almost all previous high-probability results for finding stationary points of non-convex functions establish upper bounds only for the average of norms or squared norms of the gradients along the trajectory. Moreover, Theorem 3 implies that after $\mathcal{T} \sim T^{\frac{\mathfrak{p}}{3\mathfrak{p}-2}} = o(T)$ iterations momentum $m_t$ approximates the true gradient $\nabla f(x_t)$ with accuracy $~ \frac{1}{T^{\frac{\mathfrak{p}-1}{3\mathfrak{p}-2}}}$ (which is quite good) with high probability. Intuitively, it implies that the updates become less noisy and more similar to the updates normalized GD that guarantees a decrease in the objective. This result partially explains the empirical success of the warm-up stepsize schedule.

4. **First results under second-order smoothness in such generality.** The authors generalize the results of Theorem 2 and 3 to the case when the objective function has Lipschitz Hessian in addition to the Lipschitzness of the gradients. The derived rate of convergence is $O\left(\frac{\log(T/\delta)}{T^{\frac{2\mathfrak{p}-2}{5\mathfrak{p}-3}}}\right)$ that is slightly better than the one derived without second-order smoothness. There are no analogs of these results for the convergence in expectation.

## Weaknesses

1. **Proofs are hard to read.** I have checked all the proofs and noticed a number of mathematical inaccuracies, typos, and poorly explained derivations (see the comments below). Although all of them can be easily corrected, it is vital to fix these issues.

2. **Boundedness of the gradients.** The assumption on the stochastic gradient implies boundedness of $\|F(w)\|_*$ for all $w$ that is quite strong. However, almost all high-probability results for finding stationary points in the non-convex case rely on this assumption. Therefore, this drawback is not that significant given the contribution of the paper.

## Questions and Comments

1. **line 24:** Even if the result is well-known the reference is required.

2. **line 50, $\tilde{O}()$:** The authors should either show the dependence on $\delta$ or write explicitly that $\tilde{O}()$ hides **polylogarithmical factors** of $1/(\varepsilon\delta)$.

3. **lines 89-90, about $\ell_p$-norm:** The authors should either prove this result somewhere in the appendix or add the reference to the source where it is proven.

4. **line 92, "In this paper, we will exclusively consider the case $p = 2$ and $C\ge 1$":** this statement should be either highlighted or repeated explicitly in the statements of the main result. The reason why I ask the authors to do this is that in the section on concentration bounds main results for Banach spaces are proven for general $p$. It is misleading when in the proof of Theorem 2 concentration inequalities are applied with $p=2$.

5. **line 95:** Appendix D $\to$ Appendix E.

6. **line 96, incorrect definition of smoothness:** one should use primal norm in the right-hand side of the inequality.

7. **line 98, "We assume that $\nabla f(w,z)$ satisfies ...":** In fact, the authors assume boundedness of the $\mathfrak{p}$th moment of the stochastic gradient by $G^{\mathfrak{p}}$, not the central $\mathfrak{p}$th moment.

8. **line 109:** Banac $\to$ Banach, In $\to$ in

9. **line 110, the last word:** normalization?

10. **line 111, Lemma 1:** the statement is strange, some word was omitted. Moreover, sometimes authors use $w$, sometimes -- $\vec{w}$. The notation should be unified.

11. **Algorithm 1, Input:** one should write that $\beta = 1-\alpha$ or replace $\alpha$ by $\beta$. Next, there is a typo in the update rule for $w_{t+1}$: one should have $w_{t+1} = w_t - \eta d(m_t)$.

12. **line 120** matches $\to$ matches up to logarithmical factors.

13. **line 129:** must have $\to$ must have with probability at least $1-\delta$.

14. **Formulas from Theorem 2, 3 and almost all other formulas in the paper:** there are a lot of mistakes with the punctuation after the formulas. That is, if the sentence ends with a formula, it is needed to add a full stop after the formula. Please, fix it everywhere in the paper.

15. **line 147, "By smoothness we have ..."**: it should be primal norm in the right-hand side.

16. **line 149, the inequality below:** $(1-\alpha)^t\hat\epsilon_0 \to (1-\alpha)^{t+1}\hat\epsilon_0$. Please fix it here and everywhere below. The same problem is in Theorem 3, 5, and 6 (as well as incorrect indexes).

17. **line 177, the power in the denominator:** $\frac{2\mathfrak{p}-1}{3\mathfrak{p}-2} \to \frac{\mathfrak{p}-1}{3\mathfrak{p}-2}$.

18. **Theorem 5, definition of $\eta$**: $a \to s$.

19. **Theorem 6**: a number of mistakes in the statement, that should be fixed (incorrect bounds and rates).

20. **lines 440-441 and the formula behind:** please, do not skip the steps in the derivations. It becomes hard to read the proof. Moreover, there are too many typos in these 6 lines: incorrect definition of $\epsilon_t$ (should be $x_t$ instead of $w_t$), in the formula for $m_{t+1}$ it should be $g_{t+1}^{\text{clip}}$ instead of $g_{t}^{\text{clip}}$, wrong sign in front of $\nabla^2 F(w_{t+1})$ and superfluous right round bracket after $\epsilon_{t+1}$ in the last line of the formula.

21. **line 442, the formulas behind:** the indexes in the sums are incorrect.

22. **line 443, $\|S(a,b)\|_{*} \le \frac{\rho}{2}\|a-b\|^2$**: one should add either a proof or a reference.

23. **line 443, formulas behind**: the authors skipped a lot of steps of derivations of these inequalities. It is inconvenient for readers. Moreover, the constants are not tight: one can get 2 times smaller factor in front of the sum.

24. **lines 444-445 and inequality behind, "by exactly the same argument as in Theorem 2"**: why numerical constants are different then? The authors should write a complete proof or fix the issue. In the next formula, the numerical constants differ again...

25. **Lemma 8:** It should be mentioned explicitly in the statement that this result is only for Hilbert spaces.

26. **line 483:** Lemma 14 $\to$ Lemma 12

27. **line 484:** $\hat\mu_t \to \hat X_t$.

28. **line 496, application of Lemma 14**: In Lemma 14 one needs to have bounded variance while you show only the boundedness of the central $\mathfrak{p}$th moments. Moreover, all steps in obtaining the inequality after line 497 should be added (it is not just an application of Lemma 14). Although it can be derived, I strongly encourage the authors to add all missing derivations to the proofs.

29. **Appendix C:** Feedman $\to$ Freedman

30. **lines 509 - 514, definition of $s_t$:** this definition implies that $s_1 = 0$, and, as a consequence, $s_t = 0$ for all $t$. This is because for $t=1$ the sum $\sum_{i=1}^{t-1}s_i = 0$ since it does not contain summands (it is a common convention). Therefore, formally, all the proofs in the paper are incorrect. But this problem can be easily corrected by changing the definition of $s_1$. Moreover, in the definition of $s_t$ it should be an inner product of $X_t$ and the gradient. In the current form, it is incorrect.

31. **inequality below line 518:** sums of $s_t$ and $X_t$ should be swapped.

32. **line 524, "the function $x \to x^p$ is convex...":** this assumes that $p \geq 1$. This assumption was never mentioned explicitly.

33. **lines 527-529 and inequality behind:** again, too many typos and too many derivations are omitted. First of all, from the explanation, it does not follow what is written in the inequality. To get this, the authors should change the definition of $b+c$ (take maximum over $i\leq T-1$ and upper bound it after, in the second line of the inequality). Next, in the second line of the inequality, it should be summation from $i=1$ to $T-1$ for $s_i$. To conclude, I repeat once again that the proofs are not reader-friendly and should be filled with detailed explanations. All the inaccuracies should be corrected.

34. **Lemma 11:** the reference is missing. It is not a classical form of the Freedman's inequality (for the classical one, see Theorem 1.1 from Tropp, J. (2011). Freedman's inequality for matrix martingales. Electronic Communications in Probability, 16, 262-270.).

35. **line 542:** $1-\delta \to 1-3\delta$

36. **line 544, "$|s_t| = \|X_t\|$"**: it is not true. The authors should add a simple derivation via Cauchy-Schwarz showing that $|s_t| < \|X_t\|$.

37. **line 545:** Why is it needed to use $2\delta$ instead of $\delta$ here?

38. **inequality after line 550, $R^2$ under the square root:** it should be $R^4$.

39. **inequality after line 551:** $\sigma_t^2 \to \|X_t\|^2$ in the first line, $\sigma_t \to \sigma_t^2$ in the last line.

40. **line 560:** Banach space $\to$ $(p,C)$-smooth Banach space.

41. **line 562:** $1-\delta \to 1-3\delta$

42. **expression after line 563:** What is the statement here? Is it the right-hand side of the previous inequality under additional assumptions on $p$ and $C$?

43. **line 564:** when you apply Jensen's inequality you use $p \leq 2$ implicitly. So, it should be added in the statement of the lemma even for the first bound.

44. **Proof of Proposition 16:** the proof is incorrect (though the statement of the proposition is correct).

## Comment after rebuttal
I thank the authors for their response. Since the authors properly addressed my main criticism, I am increasing my score to 8 in the hope that the authors will apply all necessary corrections in the final version of the paper.

**Time Spent Reviewing:**

12 hours

---

> ### Author Response · Authors · 2021-08-10
> **response to reviewer KPvy**
>
> Thank you very much for your detailed and thorough review! We are extremely grateful for your careful reading and comments. We will of course correct the issues you have raised, all of which are correct. As you know, many are relatively minor, so in the interest of brevity we will just address some of the points below. However, if we miss anything of concern, please do let us know during the rolling discussion period and we will update the response!
>
> (9) the sentence should end with “normalized gradient step.”
>
> (10) the lemma should say “...L-smooth *function* from…”
>
> (16) You are right - although of course this only tightens the bounds by introducing an extra factor of $(1-\alpha)\le 1$.
>
> (19) Thanks for bringing this up: as you suggest, all instances of $\frac{p}{3 p-2}$ should instead be $\frac{2p-1}{5p-3}$, and the prefactor on the burnin-in time $\mathcal{T}$ should be $T^{\frac{2p}{5p-3}} /b$.
>
> (20) You’re right, we will fix the typos and add clarifying steps.
>
> (24) The line directly after 445 has incorrect constants - the following line has the correct constants. The algebraic steps should indeed be identical to lines 154-157, but we will add the explicit calculations in the revision.
>
> (28) We are applying Lemma 14 to $\hat X_t - \mathbb{E}[\hat X_t]$, for which there is a bound on the variance in the previous lines rather than just the central moment. However, we will as you suggest add clarifications to make the proof more readable.
>
> (30) Thanks for pointing out this initialization issue, and the missing inner-product. We will correct both.
>
> (33) We think this one is correct as stated, although your suggestion would also prove the same thing. Using our definitions of $a,b,c$, we achieve the bound by writing $(a+b)^p + C\|X_t\|^p \le (a+b+c)^p + b^p - (b+c)^p + C\|X_t\|^p\le (a+b+c)^p$ since $b^p + C\|X_t\|^p\le (b+c)^p$ by definition.
>
> (34) The version we presented (note there should be a sum over the variance terms from 1 to $k$ inside the square root) can be derived from Tropp Theorem 1.1 via an application of the quadratic formula (it is actually a bit weaker than Tropp Theorem 1.1). We were remiss in not adding the reference, and we will also add the derivation after the theorem statement.
>
> (37) We are using $2\delta$ because we use union bound to convert a 1-sided concentration bound into a two-sided bound on the absolute value of the deviation.
>
> (42) Yes. The statement should say that the written expression is a simpler upper bound on the LHS of the previous expression ($\|\sum_{t=1}^T X_t\|$), under the additional assumptions that $p=2$ and $C\ge 1$, which are the conditions considered in the main text.

---

> > ### Comment · Reviewer_KPvy · 2021-08-22
> > **Thanks for the response! I am increasing my score from 7 to 8.**
> >
> > I thank the authors for their response. Since the authors properly addressed my main criticism, I am increasing my score from 7 to 8 in the hope that the authors will apply all necessary corrections in the final version of the paper.

---

### Official Review · Reviewer_WuQp · 2021-07-16

**Rating:** 7
**Confidence:** 4

**Summary:**

This paper focuses on high probability convergence behavior of nnormalized SGD with momentum and gradient clipping under heavy-tailed noise but finite moments. Specifically, without the light-tail assumption, e.g. sub-Gaussian noise, and with bounded pth moment of the gradient for $p \in (1,2]$,
-	They propose an algorithm that achieves $\mathcal O (\log(T/\delta) / T^{\frac{p-1}{3p-2}})$ convergence with high probability. They also characterize the behavior of last iterate and corresponding function value.
-	Under additional second-order smoothness, they propose a modified version of the aforementioned algorithm, which achieves convergence rate of order $\mathcal O (\log(T/\delta) / T^{\frac{2p-2}{5p-3}})$. To my knowledge, this is the first high probability results under heavy tails for second-order smooth functions.
Authors also demonstrate the effect of initial burn-in period as discussed in Theorem 3 (and Theorem 6 for second order smoothness) through some experiments.

**Limitations And Societal Impact:**

The authors don't talk about the negative societal impact of their work, but given that this is a theoretical paper it is probably less likely that it would have any immediate negative societal impact.

**Main Review:**

Prior work on high probability analysis of first-order methods usually rely on light-tail assumptions or almost-sure boundedness of gradient. This work specifically avoids those two assumptions, and achieve the same convergence rates with the in-expectation counterparts with heavy tailed gradients. Additionally, to the best of my knowledge, under heavy tails assumptions, they show high-probability convergence for second-order smooth functions.
Authors also show some complementary result on:
-	Behavior of the last iterates generated by Algorithm 1 and 2.
-	Quality of first-order estimate after burn-in period
-	Conditions under which the monotonic descent in objective is achieved

These results both apply to first and second-order smooth settings. One small criticism I have is that both Algorithm 1 and 2 require a fixed time horizon, however, it doesn’t decrease the impact of these results in my opinion.
I think the related work is adequately cited up to some relevant results from the convex realm. It might be helpful to motivate the removal of light-tails and almost-sure boundedness.

This paper is theoretically oriented, and authors provide theoretical analysis for their main theorems both in the main text and the appendix. To complete some of their theoretical findings, authors run benchmark experiments for the demonstration of burn-in idea. According to my interpretation, this burn-in period is also a parameter to be tuned, and if picked properly, it helps with the optimization process by bypassing the initial steps during which the algorithm might perform poorly. The authors are honest with their results and explain the implications of their findings with intuitions and examples, which strengthens their paper.

I believe their results are significant from a theoretical point of view as this the first result of its kind for second-order smoothness. Theorem 3 and 6 provide a detailed view for the behavior of the gradient estimate, as well as the bound on full gradient when the algorithm runs beyond a time period called burn-in. As a complementary result, Corollary 4 and 7 validate the behavior of last iterate and reinforces the practice of using the last iterate generated by the algorithm.
The paper is well-organized and has a clear presentation of the problem setting and the main results. There are some typos and incomplete sentences, which should be corrected for the final version. Authors provide explanations for their algorithmic choices and explain what each theorem implies both theoretically and in practice.

Finally, I have some particular question/comments regarding the paper:
-	Incomplete sentence at line 109
-	Could the authors please explain how they obtain the inequality right after line 173? Also, using the definitions of $\mathcal T$ and $\eta$, this quantity is upper bounded by $O ( \log(T) / T^{\frac{p-1}{3p-2}} )$, which implies a sublinearly decreasing bound. If what I computed is correct, this is an unexpected result.
-	If I didn’t miss it, G-Lipschitz assumption does not exist in the Theorem 3 but in its proof.
-	In the conclusion, it reads “Specifically, the last iterate has function value smaller than the function value of a critical point. ” I believe it should be $\epsilon$-critical point, as otherwise it might imply that $w_T$ is not near a stationary point but has smaller function value than some stationary point.



**Time Spent Reviewing:**

8

---

> ### Author Response · Authors · 2021-08-10
> **author response to reviewer WuQp**
>
> Thanks very much for your work reviewing our paper, and we are glad to hear that you find our results significant!
> In regards to the time horizon, we absolutely agree that it would be better to have a naturally decaying step-size or similar adjustment to deal with an unknown time horizon. We chose to focus on the fixed-horizon case in order to avoid introducing extra complications, but we’d like to point out that it is possible to use a variation on the doubling trick to achieve a unknown-horizon result without too much difficulty, since we are guaranteed not to increase the objective value very much over the course of any given time horizon.
>
> Line 109: The sentence should end “normalized SGD step.”
>
> Line 173: The inequality follows because each step satisfies $\|w_t-w_{t+1}\|=\eta$ so that $|F(w_t)-F(w_{t+1})|\le G\|w_t-w_{t+1}\|\le G\eta$ for all $t$. Repeating this argument $\mathcal{T}$ times yields the inequality.
> Indeed, as you notice, the bound is decreasing with $T$ for $p>1$. Although counter-intuitive at first blush, this is actually reasonable behavior: this inequality is measuring how far the algorithm travels during the “burn-in” period. Intuitively, traveling a long distance during this period would actually be detrimental since the gradient estimates are poor. Thus, we instead travel a very short distance (even though the number of iterations in the burn-in period does increase with $T$, because the learning rate decreases faster than the burn-in period increases).
>
> We will make sure the Lipschitz assumptions are clarified in the theorem statements.
>
> Yes, we mean $\epsilon$-critical point, and we will clarify in the prose.

---

### Decision · Program_Chairs · 2021-09-27

**Decision:**

Accept (Oral)

**Comment:**

The paper considers non-convex SO using first-order algorithms for which the gradient estimates may have heavy tail distributions. Using gradient clipping normalized GD with momentum is shown to convergence to near critical points in high-probability with best-known rates for smooth losses when the gradients only have bounded p'th moments for some p between 1 and 2. Next high-probability bounds are obtained for the case of second-order smooth losses.

The reviewers are unanimously strongly in favor of accepting the paper and I concur. This seems like a nice contribution.